



# Quantifying human impacts on hydrological drought using a combined modelling approach in a tropical river basin in Central Vietnam

A.B.M Firoz[1]; Alexandra Nauditt[1]; Manfred Fink[2] and Lars Ribbe[1]

[1] Institute for Technology and Resources Management in the Tropics and Subtropics (ITT), TH Köln, Cologne, 50679, Germany,
[2] Institute for Geoinformatics, Department of Geography, Friedrich-Schiller-University Jena, 07743, Germany

*Correspondence to*: A.B.M Firoz (abm.firoz@th-koeln.de)

**Abstract.** Hydrological droughts are one of the most devastating disasters in terms of economic loss in Central Vietnam and other regions of South East Asia severely affecting agricultural production and drinking water supply. Their increasing frequency and severity can be attributed to extended dry spells and increasing water abstractions for e.g. irrigation and hydropower development to meet the demand of dynamic socioeconomic development. Based on hydro-climatic data for the period from 1980 to 2013 and reservoir operation data, the impacts of recent hydropower development and other alterations of the hydrological network on downstream streamflow and drought risk were assessed for a mesoscale basin of steep topography in Central Vietnam, the Vu Gia Thu Bon (VGTB) river basin. The Just Another Modelling System (JAMS) /J2000 was calibrated for the VGTB river basin to simulate reservoir inflow and the naturalized discharge time series for the downstream gauging stations. The HEC-ResSim reservoir operation model simulated reservoir outflow from eight major hydropower stations as well as the reconstructed streamflow for the main river branches Vu Gia and Thu Bon. Drought duration, severity and frequency was analysed for different time scales for the naturalized and reconstructed streamflow by applying the daily varying threshold method.

Efficiency statistics for both models show good results. A strong impact of reservoir operation on downstream discharge at the daily, monthly, seasonal and annual scale was detected for four discharge stations relevant for downstream water allocation. In accordance with the reports from local stakeholders, we found a stronger hydrological drought risk for the anthropogenically impacted reconstructed streamflow. We conclude that the calibrated model setup provides a valuable tool to quantify the different origins of drought to support cross-sectorial water management and planning in a suitable way to be transferred to similar river basins.

## 1 Introduction

River basins and their hydrological systems play a key role in providing freshwater to downstream deltaic systems, for irrigation and domestic water supply and to regulate salt water intrusion (Ribbe et al., 2017). The patterns of timing and




magnitude of streamflow essentially depend on climatic variables such as precipitation (Souvignet et al., 2013; Min et al., 2011; Zhang et al., 2007; Ahn and Merwade, 2014), temperature and the resulting altered evapotranspiration rates (Ahn and Merwade, 2014; Santer et al., 2011; Trenberth, 2011; Vörösmarty et al., 2000), as well as on the modification of the hydrological systems by humans introducing water infrastructure such as reservoirs and damming, inter-basin water transfers

and construction of weirs. Further important influences on the water dynamics is anthropogenic water consumption for irrigated agriculture, industrial and domestic water supply (Zhou et al., 2012; Patterson et al., 2013; Ahn and Merwade, 2014; Hu et al., 2015; Wang et al., 2016; Zhou et al., 2012; Lauri et al., 2012; Rossi et al., 2009; McClelland, 2004).

Hydrological droughts are becoming more frequent disasters worldwide which can also be attributed to both hydro-climatic and anthropogenic changes (van Loon et al., 2016; AghaKouchak et al., 2015; van Lanen et al., 2016). Regional studies

show that in anthropogenically modified river basins -- in particular those altered by hydropower development and operation -- larger changes in streamflow have been observed than in hydrological systems which are only affected by climate variability and change (Ahn and Merwade, 2014; Arrigoni et al., 2010; Tang et al., 2014). Such alterations of the hydrological system often negatively affect downstream discharge patterns and communities dependent on the provision of freshwater for irrigation and domestic water supply (Zhou et al., 2012; Rossi et al., 2009; Song et al., 2015). Therefore

seasonal impacts of reservoir operation on low flow patterns and trends need to be quantified in order to separate them from natural drought propagation and to inform downstream water users to properly manage water supply for irrigation, industry and domestic water supply.

The effects of reservoir operation on streamflow have been assessed for instance in the Lena, Yenisei and Ob' river basins of the arctic Eurasian river system, on a seasonal and annual basis revealing that reservoir operation accounts for most of the

seasonal changes in the three river basins, ranging from 60% to 100% particularly in winter and early spring. Reservoir operation was found to have little effect on annual trends (Adam et al., 2007; Ye et al., 2003, 2003; Adam and Lettenmaier, 2008). (Räsänen et al., 2012) quantified hydrological changes in the upper Mekong basin due to hydropower operation in China showing an increased December–May discharge by 34–155 % and a decreased July–September discharge by 29–36 %. Impacts of the worldwide largest Three Gorges reservoir constructed in 2003 were quantified by (Zhang et al., 2015) who

assessed streamflow at three outlets on the south bank of Yangtze tributary Jingjiang River providing evidence that the reservoir impacts were largely responsible for major droughts downstream.

Positive impacts of reservoir operation on downstream hydrological regimes for example have been reported (Song et al., 2015) suggesting a decreasing frequency of flood events in the Sanchahe River Basin, China and by (Tang et al., 2014) who showed an increasing surface run-off during the dry season at the upper Mekong/Lancang River in China.

Various approaches have been used to quantify and separate anthropogenic and climate change impacts on streamflow. Most commonly used approaches, are streamflow time series analyses looking at seasonal and frequency patterns to assess impacts of human alterations on discharge. (Wang and Hejazi, 2011) used Budyko curves (Budyko, 1974), to detect human induced changes in streamflow investigating their deviation from the initial relationships between mean annual precipitation, evaporation and potential evaporation as defined by the Budyko curves. Double mass curves (DMC) are applied to compare




the cumulative distribution of precipitation and discharge time series before and after human alterations (Wang et al., 2016) as well linear regression to establish the relationship between discharge and different climatic variables (Wang et al., 2012; Sharon A. Johnson et al., 1991; Hu et al., 2015). However, although such relatively simple statistical analyses of hydro-climatic time series might give a first insight in system behaviour; they might not capture the non-linear nature of hydrological systems.

In fewer studies hydrological models have been applied to assess the different causes for streamflow changes (Zhang et al., 2012; Bao et al., 2012; Chang et al., 2015; Tesfa et al., 2014) providing simulations of naturalized and reconstructed discharge time series to quantify and separate the different impacts. Alternatively, the paired basin approach has been used to model the impact of human induced land cover changes on streamflow by comparing simulations in catchments of very similar characteristics (Seibert and McDonnell, 2010; Bonell and Bruijnzeel, 2005).

However, the above described studies had the aim to either evaluate human impacts on general streamflow behaviour or on flood risk. The implication of reservoir operation and other human alterations of the hydrological system for drought severity, duration and frequency length have not been addressed in such studies. Also hydrological drought risk is usually looked at on a monthly, seasonal, annual or long term scale. Hydro-climatic dynamics in the tropics, however, are fast and water management related decisions need to be made based on daily information eg. to avoid salt water intrusion into the irrigation and drinking water supply systems (Nauditt et al., 2017).

The overall aim of this study was therefore to quantify and separate the impact of hydropower reservoir operation and climate variability on hydrological drought in the VGTB river basin. Its specific objectives were to (1) simulate discharge for sub-basins throughout the basin as reservoir inflow and to obtain naturalized streamflow time series by applying a distributed Hydrological Response Unit (HRU) (Pfenning et al., 2009) based rainfall-runoff model J2000 ( (Fink et al., 2013; Krause, 2002); (2) model reservoir storage and operation for eight major hydropower reservoirs in order to simulate daily release rates, hydropower production and storage using the HEC-ResSim model  (USACE 2007); (3) simulate reservoir impacted reconstructed streamflow for downstream stations at the two main river branches and (4) quantify to which extent hydrological drought duration and severity can be attributed to  hydropower reservoir operation or climate variability by applying the variable threshold method approach (Tallasken et al., 1997; Sung and Chung, 2014) to reconstructed and naturalized stream flow time series.

The combined assessment approach developed in this study enables us to assess the interactions between climate, catchment and reservoir operation on the one hand and water and energy demand on the other. Furthermore it provides us with a tool to determine drought risk on a daily scale to support water management for irrigation and drinking water supply. The results of this research give a detailed insight to the current and potential impacts of reservoir operation on the downstream water availability which we provided to the water managers, the reservoir operating agencies and other decision makers..



## 1.1 Study Area: Vu Gia Thu Bon River Basin

The Vu Gia Thu Bon river basin (VGTB) is located in Central Vietnam (6o 55'-14 o55' N and 107o15'-108o 24' E), covers a total area of approximately 12,577 km2 and shares borders with the Huong River Basin (Hue Province) to the north, the Mekong River Basin (Laos) to the west and with the Tra Khuc River Basin (Quang Ngai Province) in the south. Main
provinces in the VGTB are Quang Nam and Da Nang (Fig. 1). It has a steep topography ranging from 0 m at the coast to 2,598 m of elevation in the South Truong Son Mountains in the west and by the Kon Tum mountain mass in the south. Almost half of the land area is covered by forest (47%) followed by cropland (26%) and grassland (20%) (Avitabile et al., 2016). Paddy rice cultivation and livestock farming are the two main agricultural activities in the basin. Two crops of paddy rice are planted per year in the lowlands and areas along the major rivers and yield to 5.05 tons-ha in average in 2013
(Quangnam Statistical Office, 2014)). The VGTB is home to approximately 2.5 million inhabitants (2013), 80% of which live in the coastal lowlands, 45% of which live in the urban areas (General Statistics Office, 2014). The climate in the VGTB basin is characterized by a strong rainy season lasting from September to December, which however mainly influenced by the monsoon and typhoons (Souvignet et al., 2013).. Almost 65-80% of the total annual rainfall happens during wet season, in which 70-85 % of the total rainfall occurs in October and November and is responsible for severe floods in the region
Contrast to this, VGTB experiences an extended dry season lasting from January to August and is regularly accompanied by droughts. February to April is the driest month- a period accounting for only 3-5 % of the total annual rainfall, resulting in severe water shortages and problems with saline intrusion at the coast (Souvignet et al., 2013).

The VGTB river system is formed by two major rivers, the Vu Gia and the Thu Bon, originating in the highlands, crossing the two major cities of Da Nang and Hoi An at the coast and ended into the ocean. The basin area of Vu Gia until reaching
Ai Nghia station is approximately 5,453 km$^2$, and the area of Thu Bon until Giao Thuy station is 3532 km$^2$. Around 3 km beyond the Giao Thuy station, the river enters the tide-affected area and the hydrological regime of the river behaves under the interaction of tidal and upstream inflow. At two hydrological stations - Nong Son (Thu Bon River) and Thanh My (Vu Gia River) discharge has been measured since 1976 (Fig. 1).

Water resources in the VGTB have been intensively developed for a variety of uses, including hydropower generation, large
rice irrigation systems in the delta, domestic and industrial water supply.  From 2009 until 2014, eight large hydropower reservoirs and plants have been constructed, which have a cumulative storage capacity of more than 2 km$^{3.}$ (Table 1). Inter-basin water transfer from the Vu Gia to the Thu Bon sub-basin to generate electricity from Dak Mi 4 hydropower plant is causing significant changes in the respective flow regimes. Paddy rice is still the dominant crop as it accounts for approximately 70% of irrigated agricultural area (Pedroso et al., 2016). Water stress during drought periods is a major
constraint to agricultural production in the region. Figure 2 shows mean monthly inter-annual discharge for the four gauging stations addressed in this study.



## 2 Data and Methods

### 2.1 Data

#### 2.1.1 Hydro-meteorological data

Hydro-climatic records for the basin were obtained from the Regional Center for Hydro-meteorology (RCHM). For this
study, daily data from sixteen rainfall stations, three climate stations (temperature, evaporation, humidity, radiation, sunshine
hours, solar radiation) and two discharge stations were used (see Fig. 1) for the period from 1980 to 2013, to cover an ideal
time frame (>30 year) for a most of the available stations (Souvignet et al., 2013). The two discharge stations Nong Son and
Thanh My are located in the upper basin of Thu Bon and Vu Gia river respectively (Fig. 1). These stations are not influenced
by the hydropower stations located upstream of the tributaries in the North (e.g., A Vuong, Bung 4, 5, 6 and Song Con 1, 2).
Therefore, we included two more stations (Ai Nghia and Giao Thuy) located further downstream of the basin to capture the
effect of these tributaries and hydropower reservoir operation. The latter two stations only measure water level and are
strongly influenced by tide (Giao Thuy) and tend to be flooded during the rainy season (Ai Nghia). Ai Nghia station is
located downstream of the diversion channel Quang Hue, which diverts water from the Vu Gia to Thu Bon (Fig. 1).
Although the Ministry of Natural Resources and Environment (MoNRE) provided routing rules (see Table S1) to estimate
how much water is diverted from Vu Gia to Thu Bon through Quang Hue channel, we assume that Ai Nghia station is
located upstream of the diversion of the Quang Hue channel to avoid such complexity. J2000 simulations at Ai Nghia and
Giao Thuy stations generated the naturalized flow for this study to compare the effects of reservoir operation on stream flow
changes.

#### 2.1.2 Hydropower and reservoir data

Eight large reservoirs were constructed in the Vu Gia and Thu Bon river basins between 2009 and 2014, six on the Vu Gia
river basin and two on the Thu Bon. The Dak Mi 4 (A& B) Hydropower plant was built on the Vu Gia sub-catchment, but
the water is diverted at its outflow to the Thu Bon river basin. This diversion has altered the flow pattern, and has been
detected by the two existing hydrological stations of Nong Son and Thanh My. The reservoir information is summarised in
Table 1. The classification of the reservoirs is based on the Vietnamese description of large, medium and small reservoirs
(MOIT, 2015). Reservoirs which have an installed capacity of more than 29 megawatts (MW) of energy are considered as
large hydropower plants, while the medium and smaller plants are in the range of 10 to 29 MW. The remaining plant is less
than 10 MW (PPC, 2006). Only the large scale reservoirs were considered in this study due to their potential implication on
storage. Generally, the reservoirs of medium and small scale are streamflow power stations and have minor effects on
downstream discharge. Operational rules and rule curves were collected from the technical documents of each reservoir from
the Department of Investment and Trade (DOIT) under Ministry of Investment and Trade (MOIT) of the Quang Nam
province, Vietnam (See table S2).



There are three types of reservoir operational rules: a) flood operation and related spill releases, b) dam security, which corresponds to the highest acceptable water level and c) water storage objective, for upper and lower daily storage goals and hydropower production. However, all these rules strongly depend on two distinct management seasons, namely '*Flood season'* (from 16 Sep to 31 Dec), and '*Dry Season'* (from 1 Jan to 15 Sep). During the flood season, the first considerations
are dam safety and spill discharge. If the inflow is greater than the hydropower maximum discharge capacity and water level is above the flood control zone, then the water first is diverted to its full capacity to produce hydropower and the excess water within that day will be released through spill discharge to ensure flood control. During the dry season, the guide curve will determine how the release of water from the reservoir will be managed. However, for each reservoir there is a monthly target for power to be produced, and it is further controlled by the upper and lower limits of the reservoir level. Generally, if
the water level is close to the upper limit of the guide curve, then it will maximize the energy production, otherwise if it is close to the lower limit, a limited amount of water will be released from the hydropower considering the environmental flow.

## 2.2 Methods

### 2.2.1 The combined modelling – drought assessment framework

To analyse and quantify the impacts of reservoir operation on downstream low flows and to separate them from other
impacts, longer time series for both the "pristine" as well as for the impacted period are needed. We termed them "naturalized" and reconstructed" discharge, respectively, in this study. The J2000 hydrological model  was utilized to simulate daily discharge for upstream HRU outlets of the Vu Gia Thu Bon river basin system as input streamflow time series to the reservoirs as well as to provide time series for the "naturalized" flow for the four downstream stations addressed in this study (Fig. 1). Impacts of hydropower operation on downstream low flows were assessed by using the reservoir routing
model HEC-ResSim coupled off-line to the J2000 for the VGTB river basin. Each reservoir was calibrated by HEC-ResSim individually before it was included in the integrated model. The reservoir model was run on a daily time step considering the hydropower operational rules. The output of this integrated model is referred to here as 'reconstructed streamflow'. A drought analysis was then carried out for the reconstructed (reservoir impacted) and naturalized (pristine) streamflow. Figure 3 gives an overview on the applied methods. The individual methods are described in the following sections.

### 2.2.2 JAMS/J2000 HRU based Rainfall-Runoff model

The J2000 is characterized as a physical based distributed and process-oriented model, which is suitable for simulating the hydrological process of meso- and macro-scale catchments (Fink et al., 2007; Kralisch and Krause, 2006). According to (Fink et al., 2013), It is implemented in the Just Another Modelling Framework (JAMS) framework, "which is a software framework for component-based development and application of environmental models. The model describes the
hydrological processes as encapsulated or independent process modules. These modules describe for example input data regionalization and correction, calculation of potential and actual evapotranspiration, canopy interception, soil moisture and





groundwater processes" (p. 1829). Modules are described in more detail by (Nepal et al., 2014) and in the online documentation (http://ilms.uni-jena.de/ilmswiki/index.php/Hydrological_Model_J2000). The model utilises the HRU-approach for the discretisation of the basin, consisting of an overlay of land use, soil, geology and the relief parameters topographic wetness index (Böhner et al., 2002), as well as mass balance index and solar radiation index (McCune and

5 Keon, 2002; Pfenning et al., 2009).

### 2.2.3 HEC-ResSim reservoir operation model

The Reservoir System Simulation (HEC-ResSim) software developed by the U.S. Army Corps of Engineers (USACE, 2007) allows developing simulations of single or multiple reservoirs in a hydrological network based on the available hydrological (inflow) data, the physical reservoir characteristics and the operating rules. The model is comprehensively documented in

(Klipsch and Hurst, 2013). J2000 simulated inflow time series (compare locations in Fig. 4a) were introduced and routed, with reservoirs altering the routed flow based on physical constraints and operating rules (Fig. 4b). The program uses a rule-based approach to govern reservoir release, from which hydropower can be generated. Reservoirs are divided in vertical zones having rules associated with each, and the total storage is determined by a storage-elevation-area relationship. In Figure 4a and Figure 4b, the transfer points are shown as well as the node based HEC-ResSim network with the modelled

hydropower reservoirs.

### 2.2.4 Hydrological Drought Assessment

Threshold approach (Tallaksen et al., 2009), is being extensively used to determine the hydrological drought in temperate regions, where the discharge is usually greater than zero ( Tallaksen et al., 2009; van Huijgevoort et al., 2012; van Loon and van Lanen, 2012; Sung and Chung, 2014). This method however also used globally, and also in the tropical region as well

(van Lanen et al., 2013). It defines the drought event based on a threshold level and providing information of the onset, duration and its severity (Hisdal et al., 2004). For identifying a drought event, parameters of the flow below threshold ($Q_0$), drought duration $D_i$, deficit volume or severity, $S_i$, and time of occurrence $t_c$ are used for statistical characterization of a drought event (Stahl, 2011). To eliminate minor and mutually dependent droughts from the record of events, pooling procedures have been applied (R1) (Tallasken et al., 1997). According to Sung and Chung (2014), this pooling procedure is

explained as, "If the "inter-event" time $t_c$ between two droughts of duration $d_i$ and $d_{i+1}$ and severity $s_i$ and $s_{i+1}$, respectively, are less than the predefined critical duration $t_c$ and the pre-allowed inter-event excess volume $Z_c$, then the mutually dependent drought events are pooled to form a drought event (Zelenhasić and Salvai, 1987; Tallasken et al., 1997)" (p. 3343)

$$d_{pool} = d_i + d_{i+1} + t_c$$

$$S_{pool} = s_i + s_{i+1} - z_c \qquad\qquad\qquad\qquad\qquad \text{(R1)}$$



In this study we consider that:

$t_c$ =3 days and $z_c$ =10% of $d_i$ or $d_{i+1}$

The daily variable threshold approach (Hisdal et al., 2004) based on flow duration curves (FDC) has been applied to determine the droughts. In this study we have used the $90^{th}$ percentile ($Q_{90}$) of the FDC as the daily variable threshold which is obtained from the antecedent 365 daily streamflow values. $Q_{90}$ is defined as follows: for a given day of the hydrological year d (the start of the hydrological year is "01/07" for the VGTB basin), the daily varying $Q_{90}$ (d) is calculated based on moving average (MA) of 30 days centered on day d (i.e., 15 days either side), starting from the first day of the hydrological year (Prudhomme et al., 2011). As the study region has strong seasonality between dry and wet season (i.e., the distinct differences of flow in dry and wet season, see fig.2), we further introduce the break-days concept to address the seasonality, which calculate separately the threshold level for both dry and wet season. In our case, the break-days refer to the 01/07 and 01/01, which is the starting of the wet seasons & dry season, respectively. The advantage of this approach is that it can detect the deviations of the streamflow for both the dry and wet seasons. Furthermore, lower than the average flow in wet seasons could have lead to the development of drought for the following season (Sung and Chung, 2014). A binary based approach (1 or 0) has been considered to identify whether it is a dry day or normal day based on the daily low flow varying threshold. For each streamflow record, the value is compared with threshold, and if the value is less than or equal to the threshold level then it considered as dry day and replaced by a single index equal to 1 and otherwise it considered as 0 (R2) (Prudhomme et al., 2011). Finally, the streamflow deficit of the naturalized and the reconstructed streamflow are compared to quantify the impact of reservoirs on streamflow drought.

$$DI(t)Z \begin{cases} 1\ if & Q(t) \leq threshold \\ 0\ if & Q(t) > threshold \end{cases} \tag{R2}$$

## 3. Results

### 3.1. J2000 Hydrological model calibration to simulate reservoir inflow and naturalized discharge

The J2000 model was calibrated for the gauging station Nong Son for the period of 1996-2000. The calibration was conducted manually. We also performed an automatic calibration using the multi objective NSGA2 algorithm (Deb et al., 2002) which gave us similar results according to the objective functions (Table 2). We decided to use the manually calibrated parameters because of the better shape of the hydrograph with the manually calibrated parameter set. The second available gauging station (Thanh My) was not separately calibrated, but tested using the same parameter set acquired from Nong Son. This was done because we wanted to check the ability of the model to simulate the runoff for even those parts of the basin where no calibration was possible due to the lack of discharge data. Table 2 shows the results for the objective





functions for the calibration and validation period (2000-2005). Since the Vu Gia datawere not calibrated, both periods are the validation for the parameter transfer. The presented efficiencies are the (1) coefficient of determination ($R^2$) to show the goodness of fit for the general model dynamics, the (2) Nash-Sutcliffe (E2) efficiency to judge the goodness of fit with a focus on peak flow and simulated volumes and (3) the Nash-Sutcliffe efficiency (logE2) with logarithmic values to achieve a

5    stronger focus on the low flow periods (Krause et al., 2005). As an indicator for the overall simulated volumes, we used the percent bias (Pbias) (Table 2). The objective functions logE2 and $R^2$ show that the low flow periods and the overall dynamics are well represented. Because of the HRU concept of J2000 in the JAMS modelling framework, it is possible to generate hydrological state variables for each point in time and space. This facilitates the transfer of runoff data at daily basis at selected points along the river segments to the reservoir model (Fink et al., 2013).

### 3.2. Reservoir Modelling under varying operational rules

We applied the HEC-ResSim Reservoir system simulation model to simulate reservoir release, hydropower production and storage in the individual reservoirs of the VGTB at a daily time step. Based on the technical document provided by the MOIT (more detail provided in the supplementary files for the operational rules of individual reservoirs Fig. S1), the eight

reservoirs were first modelled individually, calibrated and evaluated based on the observed outflow at their outlets. The water release has been designed to reproduce the observed mean of the daily release. This ensures that the individual reservoir will reproduce well enough to simulate outflow and can then be incorporated into the integrated reservoir model. For this study, we have used the hydropower release data of four of the eight reservoirs - A Vuong, Dak Mi 4, Song Con 2 and Song Tranh 2, for which the outlet data of the turbine discharge is available. Three of the remaining four reservoirs have

only been operational since 2013 (Song Bung 4, 5 & 6), and the data was not available. The final reservoir, Song Con 1, is considered as a run-off reservoir and, therefore was not necessary to account for its outflow in this study.

The simulation time varied for each of the reservoirs, depending on their year of construction and availability of the discharge data from the turbine. For A Vuong reservoir, the available data was from February 2009 until August, 2012, and was used to compare the simulated cumulated daily release from the turbine to the observed discharge (Fig. 5). The

simulated cumulative discharge showed very good agreement with the observed discharge data from the turbine. The simulated outflow of Dak Mi 4 was in good agreement with the observed flow from January 2012 to end of December 2012. During the summer time in 2013, the simulated cumulative discharge was underestimated compare with the observed discharge (Fig. 5). This phenomenon can be explained by the designed environmental release from the Dak Mi 4 reservoir to the Vu Gia River. As mentioned above (see Fig.1), the Dak Mi 4 hydropower (located at the Vu Gia river upstream) releases

water to the Thu Bon river through turbine discharge. Therefore, to maintain the environmental flow to the downstream of Vu Gia river, the reservoir should release a minimum of 25 $m^3s^{-1}$ water from the reservoir to the Vu Gia river (MOIT, 2011). However, because of the high demand for energy during the dry season, some of the water intended for environmental flow for the Vu Gia river was used for the energy production and discharge to the Thu Bon river. The simulation result of Song



Tranh 2, was not quite satisfactory after January 2012 (Fig. 5). This is due to the fact that this reservoir experienced leakages from its dam, and any storage of water for the year 2012-2013 was prohibited due to dam safety. Any water coming inside the dam was used immediately through the turbine, increasing discharge from the turbine. As a result, there was no storage functionality in the reservoir during this period. After 2013, the leakages were repaired and the reservoir returned to its normal operating condition.

## 3.3. Simulation of reconstructed Streamflow

In this study, reconstructed streamflow was simulated which accounts for the effects of reservoirs. The calibrated individual reservoir operating rules and other physical parameter information were incorporated in the integrated modelling system which simulates the reconstructed streamflow for the period 1980-2013 with varying management options such as cascade reservoir operation, flood control and management of water for dry season. This provides the estimated streamflow at the two existing gauging stations (Nong Son and Thanh My) and the two additional locations further downstream of the mouth of the two reaches (Ai Nghia and Giao Thuy) to capture the influences of reservoirs located further downstream (Fig.1). As there are no gauging stations at Ai Nghia and Giao Thuy, we considered the output of the J2000 simulation as a reference value for the natural streamflow, as it produces robust results (described in Section 2.1.1). To evaluate the reservoir simulations, we applied the performance statistics for the period of 2011 to the end of 2013 (Table 3). This timeframe was chosen because the Dak Mi 4 and Song Tranh 2 hydropower plants started their operation after 2011 affecting the discharge stations Nong Son and Thanh My. The streamflow efficiency statistics E2, logE2 and $R^2$ (Table 3) show that the model has a good overall performance for Nong Son Station and a slightly weaker performance for Thanh My station (E2 = 0.74). This coincides with the results for rainfall-runoff modelling (Fink et al, 2013). The Vu Gia river (at Thanh My station) – which mainly supplies water for the city of Da Nang and large irrigation areas - is strongly impacted by Dak Mi 4 operation and downstream water management is strongly dependent on decisions related to hydropower production and related releases (Figure 5). Further investigation regarding the management strategies for this hydropower plant could improve the simulation performance. However, the high efficiency values (Table 3) of the simulated actual water release confirm the overall model ability to simulate the daily release and reproduce the reconstructed streamflow.

## 3.4. Simulated long term effects of reservoirs on streamflow

The solid black circles in Figure 1 show the locations of gauging stations used to compare our reconstructed streamflow under reservoir influences, with data inferred from observed streamflow. We assumed in our study that all eight reservoirs came into operation in 1980, then ran the reservoir model to produce the synthetic streamflow termed here as reconstructed flow. This gave us the opportunity to evaluate the influences of the reservoirs on long-term streamflow pattern. Raster based visual representation of the naturalized and reconstructed streamflow at a daily basis is presented in Figure 6. For Thanh My station, the effect of the diversion in Dak Mi 4 from Vu Gia to Thu Bon is clearly visible. For downstream Ai Nghia station (also affected by the diversion of Dak Mi 4) extremely low values (pink) are less frequently appearing in the reconstructed



simulations compared to the naturalized ones. This can be attributed to the damping effect of the additional reservoirs belonging to this lower part of the sub-basin and their energy production during the naturally very low flow situations. The following analyses of longer time periods show that the overall water availability in Ai Nghia is also reduced due to the diversion. For Nong Son and Giao Thuy a higher water availability is shown due to the additional water diverted to Thu Bon at Dak Mi 4.

To quantify the reservoir effects on a monthly scale, we plotted the mean monthly values of the reconstructed streamflow against the naturalized discharges for the four stations (Nong Son, Thanh My, Ai Nighia and Giao Thuy (Fig. 7). At Nong Son station, monthly streamflow had increased by 24-62 $m^3 s^{-1}$ (about 23-85% of the observed discharge) during January to August. Although the mean discharge for September to December had increased by 50 -114 $m^3 s^{-1}$, the proportion in terms of percentage was rather low varying from 1.3% in October to 26.31% in December (Fig. 7a). A sharp contrast was observed for Thanh My station located at the upstream of the Vu Gia River. Monthly streamflow was reduced on average by around 51 $m^3 s^{-1}$ (38 % of the observed flow). The impact of reservoir operation is most obvious during the months from January to August, in which it experiences a decreasing flow ranging from 30 to 60% compared to its original state. During September to December, the flow is also reduced on average by 30%. Giao Thuy and Ai Nghia stations are located approximately 25 km and 32 km downstream of the Nong Son and Ai Nghia stations respectively and exhibit a similar pattern of flow changes due to reservoir construction. The combined effect of reservoir operation over the VGTB shows that overall, the flow had increased from January to August (Fig. 7B) and a particularly significant increase of flow augmentation was found during the month of March and April (62-68%), indicating that irrigation during this time can be ensured, when the peak demand for water is required for all activities.

The flow during the rainy season decreased by -2 to -38%, which could improve the flood protection in the region. We also addressed the influences of reservoirs on seasonality of discharge. This analysis of seasonality provides insight into the changes that underlie annual effects, and provides an opportunity to assess the adequacy of data reconstruction. The reservoir operation significantly changed the seasonality of discharge from the two major rivers, Vu Gia and Thu Bon. These changes are mainly due to the construction of the Dak Mi 4 reservoir which diverts water from the Vu Gia (Thanh My station) to the Thu Bon (Nong Son station) (Fig. 8). Streamflow for the Nong Son station in both dry and wet seasons has increased significantly, but at the same time, the peak has reduced in November and December, when floods usually occurred. This indicates that normal floods in Thu Bon sub-basin have reduced, and may not have the same impact as they did before the construction of the reservoirs. For the Vu Gia side at Thanh My station, there is an overall decreasing pattern for both seasons, except the flood peak which is high during November and December. This is due to the Dak Mi 4 reservoir's spill operation which allows excess water to pass through the spillway gate.

Annual time series and mean monthly flows for each of the reconstructed products and the naturalized data are shown in Figures 8b and 8c respectively. Overall, the figures show an increasing flow at Nong Son and Giao Thuy stations, and the opposite pattern of decreasing flow for the Thanh My and Ai Nghia stations. For Nong Son and Thanh My stations, the differences are more obvious.



### 3.5. Impacts of Reservoir operation on hydrological drought

For the VGTB, streamflow drought was determined through the seasonality (Dry and Wet) along with the varying threshold level method ($Q_{90}$). Results show that low flows generally occur in spring (MAM) and extend towards summer (JJA) time (Fig. 9). Hydrological droughts were recorded for the years 1982, 1983, 1988, 1990, 1998, 2005, 2012 and 2013 (Nauditt et al., 2017). Figure 9 shows the drought onset and duration of the naturalized and the reconstructed streamflow time series to evaluate the reservoir operation impact on hydrological drought. At Nong Son station located at the upper Thu Bon river, the analysis shows a general shift of the occurrence of drought from summer to spring due to reservoir construction and operation (Fig. 9). Specifically, at the beginning of the summer (June and July) this station merely experienced streamflow deficiency which is evident in late summer (August) or the beginning of fall (September). This phenomenon can be explained if we consider two issues: Firstly, there is a higher demand of energy during summer and hydropower has to release water for energy production. Secondly, before autumn, reservoirs need to release water to create capacity to store the monsoon water. Nong Son (upper Thu Bon) and Giao Thuy (lower Thu Bon river) stations exhibit a decreasing number of drought days respectively, from 821 to 680 and from 1025 to 713 days, due to the diversion of the Dak Mi 4 reservoir from Vu Gia to Thu Bon. On the other hand, Thanh My station which is located at the upper Vu Gia, shows more days under drought for the reconstructed period (1061 days) compared to the naturalized period (774 days). Similarly, an increasing number of drought days and frequency was found at Ai Nghia from 1011 to 1286 days. The number of drought days correspond to year at each of the stations are presented in the supplement (Figure S3).

### 4. Discussion

#### 4.1 Is the combined modelling framework suitable to assess the hydrological regime under either natural or reservoir operation impacted conditions at different time scales?

Comparing observed, simulated reconstructed and naturalized discharge time series is a widely used method to successfully assess and quantify anthropogenic impacts on streamflow (Zhang et al., 2012; Deitch et al., 2013; Lopez-Moreno et al., 2014; Chang et al., 2015). Here we used the reconstructed streamflow to assess the simulation of longer reservoir impacted discharge time series to assess the long and middle term effects of hydropower operation on the one hand and seasonality and different time scales on the other. The naturalized streamflow is used to quantify the differences under exactly the same climatic conditions. (Adam et al., 2007) for example evaluated the potential contribution of artificial reservoirs to long-term changes in annual and seasonal streamflow. Similarly to our study, they used a combined distributed hydrological model together with a reservoir routing model to generate reconstructed streamflow to be compared to the pristine environment. The soft linked model setup shows very good results in terms of statistical efficiency performances and provides reliable simulations for both reconstructed and naturalized streamflow. This includes the case for the low flow simulations and hydrological drought periods which usually pose the greatest challenges to hydrological modelling (Pilgrim et al., 1988; Nicolle et al., 2014).





As time series for either the pristine or the human impacted period in most cases are too short to fulfil statistical significance, this method is considered the most reliable when assessing the short, middle and long term changes in anthropogenically impacted discharge. Also, as shown in (Nauditt et al., 2017) varying basin characteristics, such as land cover changes and basin storage only play a minor role in runoff generation processes in the VGTB basin, which is instead dominated by

5 precipitation inputs. Therefore, it can be assumed that all the quantified changes in this study for the different temporal scales can be considered as net values for anthropogenic impacts on low flow discharge.

The main disadvantage of this methodology involving several models is the fact that it is data intensive and time consuming. Also, to make the results more reliable, uncertainties especially related to reservoir operation, precipitation input data or observed discharges used for calibration and validation, need to be further evaluated, described and communicated to the

10 stakeholders.

## 4.2 Which potential uncertainties need to be addressed in this combined modelling approach?

Although the results of this study are plausible and provided reasonable quantitative information for water resources management and reservoir operation, uncertainties in the simulations have not been quantified or estimated due to the complexity of the study. To use the final model setup and resulting simulations for water management, the main sources for

uncertainties need to be identified while their relative portion contributing to the final simulations should be quantified and communicated to stakeholders.

Different sources of uncertainties in environmental modelling are described e.g. in (Walker et al., 2003; Refsgaard et al., 2007; Refsgaard et al., 2006; Refsgaard et al., 2007; Beven and Binley, 1992). For this study, uncertainties related to the following three categories are discussed: 1) data input as precipitation data, discharge and 2) operational rules of reservoirs

and 3) parameter and model setup related uncertainties.

Regarding 1), to date there is no way of reliably estimating **areal basin precipitation** especially for a high temporal and spatial resolution. More rainfall measurements in the basin headwaters would help however, this is impractical given the remoteness and inaccessibility. Therefore resulting uncertainties for hydrological modelling related to areal precipitation is a first order limitation. (Refsgaard et al., 2006) found that the uncertainties related to rainfall estimations were by far larger

than those related to parameter uncertainty. However, this depends on basin size, response time, the model type and the assumptions made in representing the different sources of uncertainty. In the data sparse VGTB river basin, uncertainties related to precipitation model inputs are very high. All climate stations are all located below 120 m of elevation and no information is available for the higher elevations of up to 2250 m.

To overcome such underrepresentation of high elevation P and missing spatial coverage of P, J2000 relies on a

30 comprehensive regionalization methodology to distribute precipitation which is described in (Krause, 2002) that combines a linear regression approach applied to station data with an elevation gradient to account for the vertical variability. The residues from the individual stations to the calculated regression line are interpolated with the Inverse Distance Weights (IDW) method to account for the lateral variability. $R^2$ of the regression line is used to determine if the relation between the





rain and the altitude is strong enough to be used for the modelling. A threshold value of $R^2$ of 0.75 is typically used for this decision. If the threshold is not met by the data, only the IDW interpolation without the altitude regression is utilized to determine the values for the individual HRUs. This combined method considers the altitude effect if it can be identified using the measured values to potentially compensate for some of the missing precipitation at higher elevations. However,

total distributed daily inputs are still uncertain and difficult to quantify.

Discharge is monitored at Nong Son and Thanh My station by measuring water level using a boat at each point of the river every day. Discharge is then calculated based on a monthly updated rating curve. This method is not very accurate especially during high flows and hence also a source of uncertainties.

Further input data contributing to output related uncertainties are the operational rules for the reservoir operation which are

strongly dependent on the energy production demand on National level. Although they should follow the rules used for calibration of the HEC-ResSim model which consider flood protection storage and a minimum release for downstream water users, real operation is much more variable on an hourly and even on a daily scale. Therefore uncertainties related to the input data **"operational rules" 2)** are more difficult to analyse as they depend on human decision making which is not always "ruled". (Mateus and Tullos, 2016) estimated changes in streamflow and reservoir reliability under climate variability

in two basins with different hydro-geological settings in North-western United States. They assessed climate related sensitivity and uncertainties in HEC-ResSim modelling performance by comparing model results with inputs from VIC hydrological modelling simulations forced with eight climate model projections. A range of output percentiles 2.5, 50, and 97.5 was assessed. They found that changes in streamflow are much more sensitive to reservoir operational rules compared to climate variability which in their case would lead to higher streamflow in spring and lower streamflow in summer. This

coincides with our observations in central Vietnam. For our study, the uncertainty range will be assessed by comparing our reconstructed streamflow with simulation results incorporating the historical real reservoir operation, with parameters such as volume and discharge differences quantified as uncertainty ranges.

**3) Model parameter and structural uncertainty** is related to the simplifying assumptions of the conceptualisation in

modelling processes compared to reality and the associated parameter values and ranges. The combined modelling framework used in this study involves a large number of such uncertainty sources belonging to each model structure and the correspondent parameters. Uncertainties related to HEC-ResSim modelling algorithms deal with channel routing and reservoir storage without taking into consideration spatially distributed environmental processes. It can be assumed that uncertainties are more related to input data and the extreme tropical hydro-morphological dynamics (Mateus and Tullos,

2016).

For the J2000 model, however, the knowledge on uncertainties related to model structure and especially parameter sensitivity would help to define an uncertainty range for the use of the model framework for predictions and decision support. The J2000 model was calibrated against the Nong Son discharge station daily time series (1996-2000) in the VGTB. The good calibration results in terms of statistical efficiencies and hydrograph representation have encouraged us to use the



model for this assessment without carrying out a comprehensive uncertainty analysis addressing uncertainties related to parameter behaviour, climate inputs, spatial data inputs and the discharge time series used for calibration and validation. The parameter range behaviour might be further analysed to identify the most sensitive features and their representation in the real environment. J2000 parameter sensitivity for selected parameters was for instance assessed by (Nepal et al., 2014) for a

5 snow-melt dominated environment. A toolbox with different statistical methods to analyse parameter sensitivity and uncertainties in J2000 was described in (Fischer et al., 2012). However, the assessment of parameter sensitivity and identifiability for 34 parameters for such a distributed model would require a large computation effort. Such potential uncertainties should be communicated to the stakeholders before using the model framework for decision support. Uncertainties in water management are often still not sufficiently addressed which can lead to that wrong decisions being

made and planning not carried out in a sustainable way.

## 5. Conclusion

The tropical mountainous VGTB Basin in Central Vietnam is frequently affected by hydrological drought and related salt water intrusion. To a large extent this is attributable to increasing water abstractions for irrigation and domestic water supply as well as to hydropower development. Based on a coupled modelling approach, the impacts of recent hydropower

development on downstream streamflow and drought risk were assessed for this mesoscale basin. To quantify such impacts for different time scales, a naturalized and a reconstructed simulated time series was needed to be compared with the observed time series. The J2000 model was used to simulate reservoir inflow discharge and the naturalized flow for the target points. The HEC-ResSim reservoir operation model simulated following on the discharge from all major existing and planned hydropower stations in the basin as well as the reconstructed streamflow for the main river branches Vu Gia and

Thu Bon. Naturalized and reservoir impacted drought risk was assessed by applying a threshold based drought severity analysis.

Efficiency statistics for both models show good model performance. A strong impact of reservoir operation on downstream discharge at the daily, monthly, seasonal and annual scale was detected for four discharge stations relevant for water allocation points of the river basin. In accordance with the reports from local stakeholders, we found a stronger hydrological

drought risk for the reconstructed streamflow at Thanh My and Ai Nghia station, located in the upstream of Da Nang city and large rice irrigation systems facing water shortage in the dry season. Uncertainties related to climatic and reservoir input data, models and parameters were not yet quantified but we observed a strong source of uncertainties related to the reservoir operational rules which were not harmonized with the designed dimensions of the reservoirs which leads to model errors. Incorporating the adapted dimension in turn allowed quantifying the volume differences. Further major uncertainties could

be identified for the precipitation input data especially for the Vu Gia sub-basin were the measurement station density is extremely low.



We conclude that the calibrated model setup provides a valuable tool to support cross-sectorial water management and planning in the region suitable to be transferred to similar regions. The threshold level method proved to be an efficient method to estimate the drought in the tropical monsoon dominated area, which however have strong seasonal characteristics. After a thorough uncertainty analysis and the development of scenarios under changing climatic and water demand

conditions, a drought management plan for the region will be developed.

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





| Item | Unit | A Vuong | Song Tranh 2 | Dak Mi 4 A | Dak Mi 4 B | Song Bung 4 | Song Bung 5 | Song Bung 6 | Song Con 2 |
|---|---|---|---|---|---|---|---|---|---|
| First year of operation | Year | 2008 | 2011 | 2011 | 2011 | 2015 | 2014 | 2014 | 2009 |
| River System | | VuGia | ThuBon | Vu Gia | Vu Gia | VuGia | VuGia | VuGia | VuGia |
| Catchment Area | km² | 682 | 1100 | 1125 | 29 | 1448 | 2369 | 2386 | 250.1 |
| Mean Annual Flow | m³ s⁻¹ | 39.8 | 106 | 67.80 | 1.1 | 73.7 | 118 | 119 | 13.2 |
| Full Supply Level (FSL) | m.a.s.l | 380 | 175 | 258 | 106 | 222.5 | 60 | 31.8 | 275 |
| Minimum Operation level (MOL) | m.a.s.l | 340 | 138 | 240 | 105 | 195 | 58.5 | 31.8 | 274 |
| Reservoir Area at FSL | km² | 9.1 | 21.5 | 10.4 | 0.45 | 15.65 | 1.68 | 0.398 | 0.13 |
| Reservoir Area at MOL | km² | 4.3 | 9.3 | 7 | 0.4 | 7.8 | 1.68 | 0.398 | 0.12 |
| Reservoir Total Storage | 10⁶ m³ | 343.6 | 733.4 | 310 | 2.6 | 510.8 | 20.27 | 3.29 | 1.2 |
| Reservoir Active Storage | 10⁶ m³ | 266.5 | 521.1 | 158 | 0.6 | 233.99 | 17.82 | 3.29 | 0.7 |
| Spillway Design Flood | m³s⁻¹ | 5730 | 11069 | 7864 | 642 | 15427 | 16780 | 17011 | 3217 |
| Maximum Tail Water Level | m.a.s.l | 86.6 | 87.5 | 108 | 71.5 | 121.3 | 32.33 | 15.5 | 29.7 |
| Normal Tail Water level | m.a.s.l | 58 | 71 | 106 | 67.5 | 101.6 | 30.7 | 12 | 18 |
| Design Head | m | 300 | 88.3 | 135 | 37.5 | 112.4 | 27 | 13.4 | 246 |
| Total Turbine Design Discharge | m³ s⁻¹ | 78.4 | 209.7 | 121 | 122 | 172.7 | 239.24 | 243.2 | 22.8 |
| Installed Capacity | MW | 210 | 162 | 141 | 39 | 156 | 57 | 29 | 46 |
| Annual Average Energy Potential | GWh | 825 | 620.7 | 582 | 161 | 618 | 220 | 151 | 168 |

Table 1: Reservoirs in the VGTB River basin (ICEM, 2008)



| Station | Thu Bon (Nong Son) | | Vu Gia (Thanh My) |
|---|---|---|---|
| Time | Calibration | Validation | Validation |
| Frame | 01.11.1996 – 31.10.2000 | 01.11.2000 – 31.10.2005 | 01.11.1996 – 31.10.2005 |
| E2 | 0.856 | 0.869 | 0.610 |
| logE2 | 0.863 | 0.856 | 0.776 |
| $R^2$ | 0.869 | 0.870 | 0.774 |
| Pbias | -10.6 | -5.37 | 8.59 |

Table 2: Performance of efficiency statistics for the J2000 hydrological model: E2, Nash-Sutcliffe efficiency; logE2 Nash-Sutcliffe efficiency with logarithmic values; R², coefficient of determination and Pbias relative volume error in percent.



| Stations | Nong Son | Thanh My |
|---|---|---|
|  | 01.01.2011-31.12.2013 | 01.01.2011-31.12.2013 |
| E2 | 0.907 | 0.716 |
| logE2 | 0.79 | 0.74 |
| $R^2$ | 0.954 | 0.809 |
| Pbias | 0.0052 | -0.077 |

Table 3: Performance statistics: Nash Sutcliff Efficiency, (E2), logE2, R², & Pbias of the reservoir model for the two gauging stations





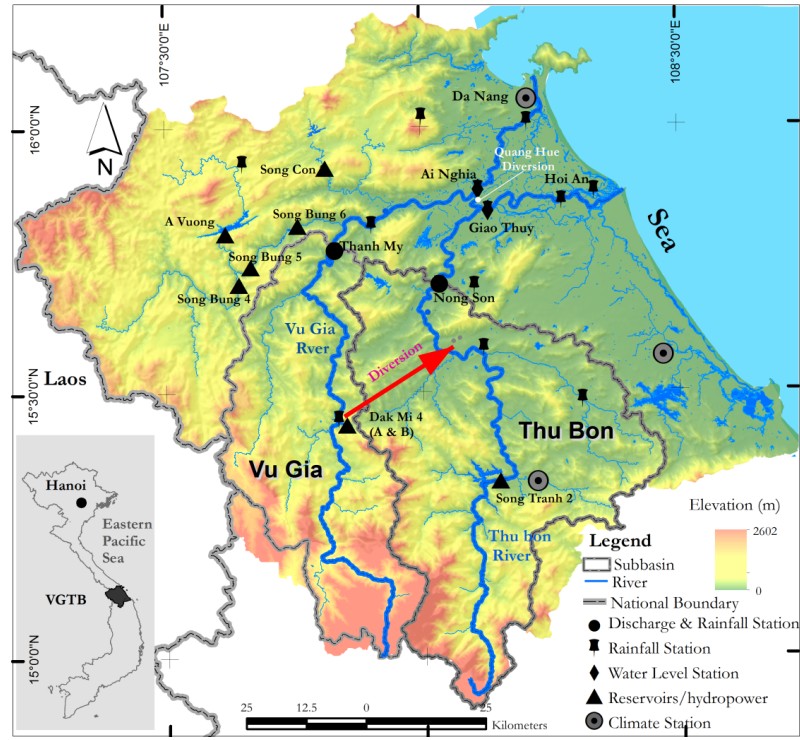

Figure 1: Topographical map of the VGTB river basin showing hydrology, hydro-meteorological monitoring network and eight major hydropower reservoirs as well the diversion (in red color) from VuGia to Thu Bon at Dak Mi 4 hydropower plant


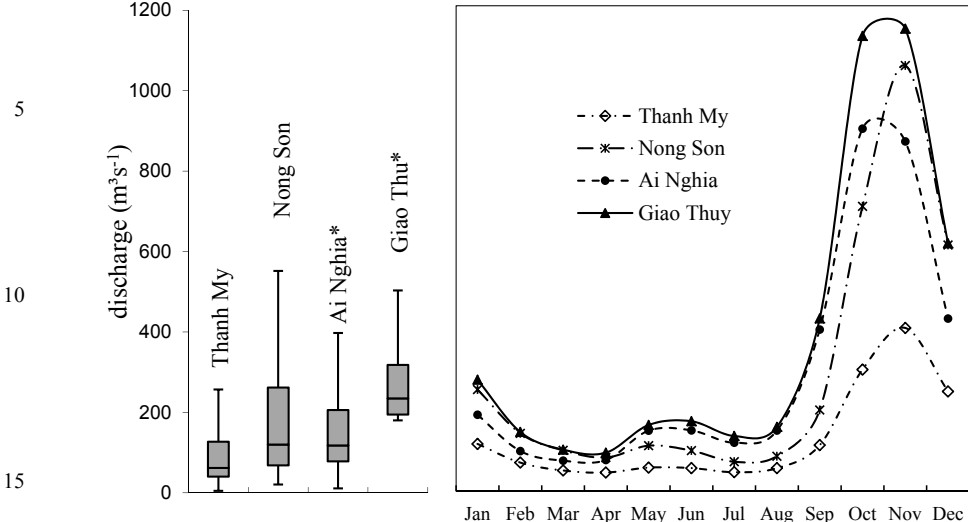

Figure 2: Mean monthly discharge at the four stations under study for the period 1979-2013 (right side). Naturalized flow data for Ai Nghia and Giao Thuy stations were simulated with J2000. Box plots (left side) indicate the 25%ile, 50%ile (median) and 75%ile of the daily streamflow time series. Outliers have been removed from the plots. The whiskers are defined as the first quartile minus 1.5*IQR and the third quartile plus 1.5*IQR.





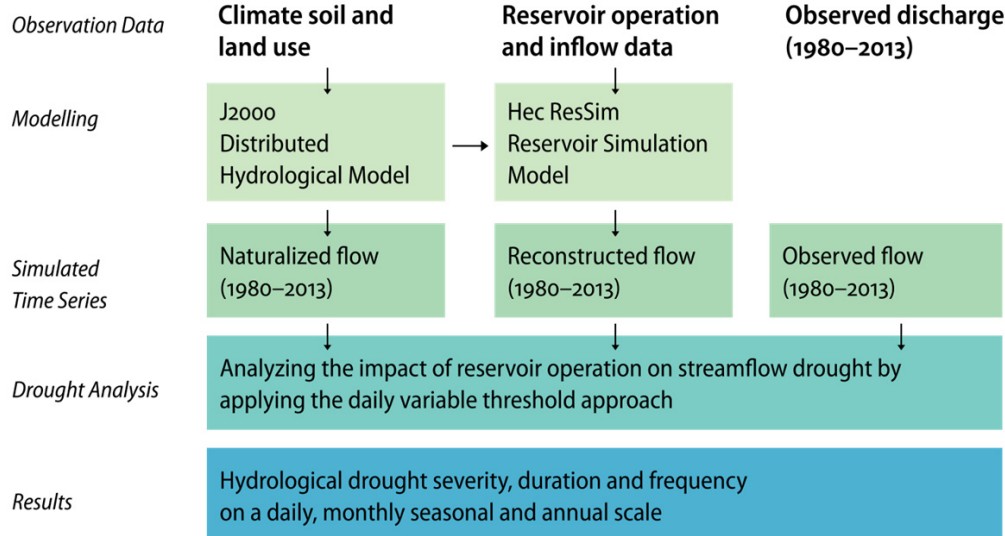

Figure 3: Drought assessment framework- 1) Distributed Hydrological Model (J2000) (Krause, 2002) provides the simulated inflow data at various nodes and naturalized streamflow 2) HEC-ResSim simulates reconstructed streamflow for the entire observation period and 3) Streamflow Deficiency analysis through threshold level methods provides information about the drought duration and extent on different temporal pattern. And the reservoir impact on the downstream streamflow has been assessed based on the reconstructed and naturalized streamflow differences.





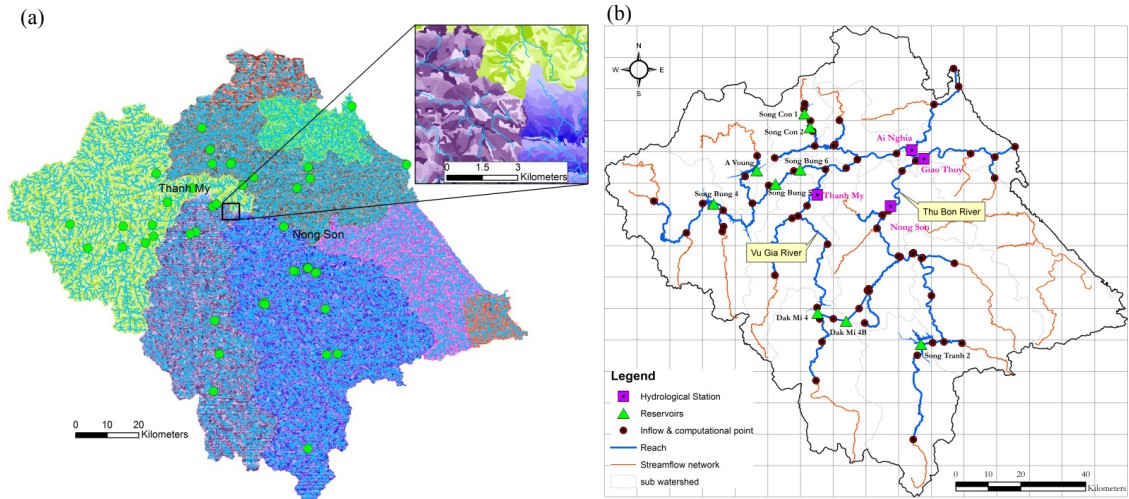

Figure 4: Coupling of J2000 model with the HEC-ResSim model. (a) HRU of the J2000 model along the major sub-basin,
virtual discharge stations (green points) for which J2000 simulated time series for the reservoir inflow and relevant
abstraction points in the downstream area. (b) The HEC-ResSim model node network, J2000 inflow discharge points (brown
dots) and the location of the reservoir that has been incorporate within the reservoir model.





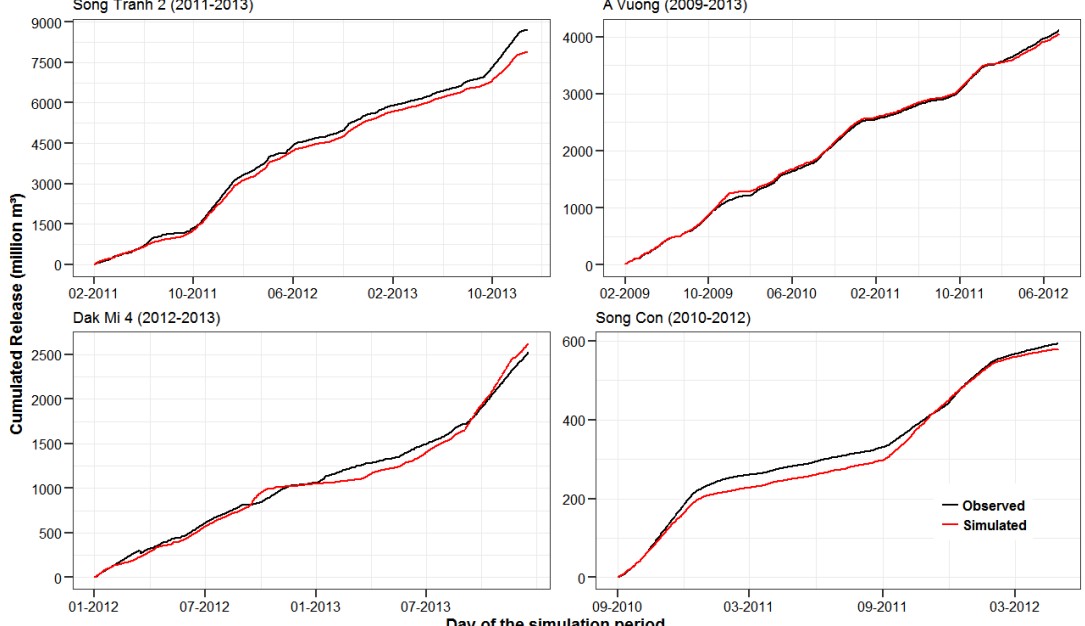

Figure 5: Simulated and observed cumulated daily release of the individual reservoirs





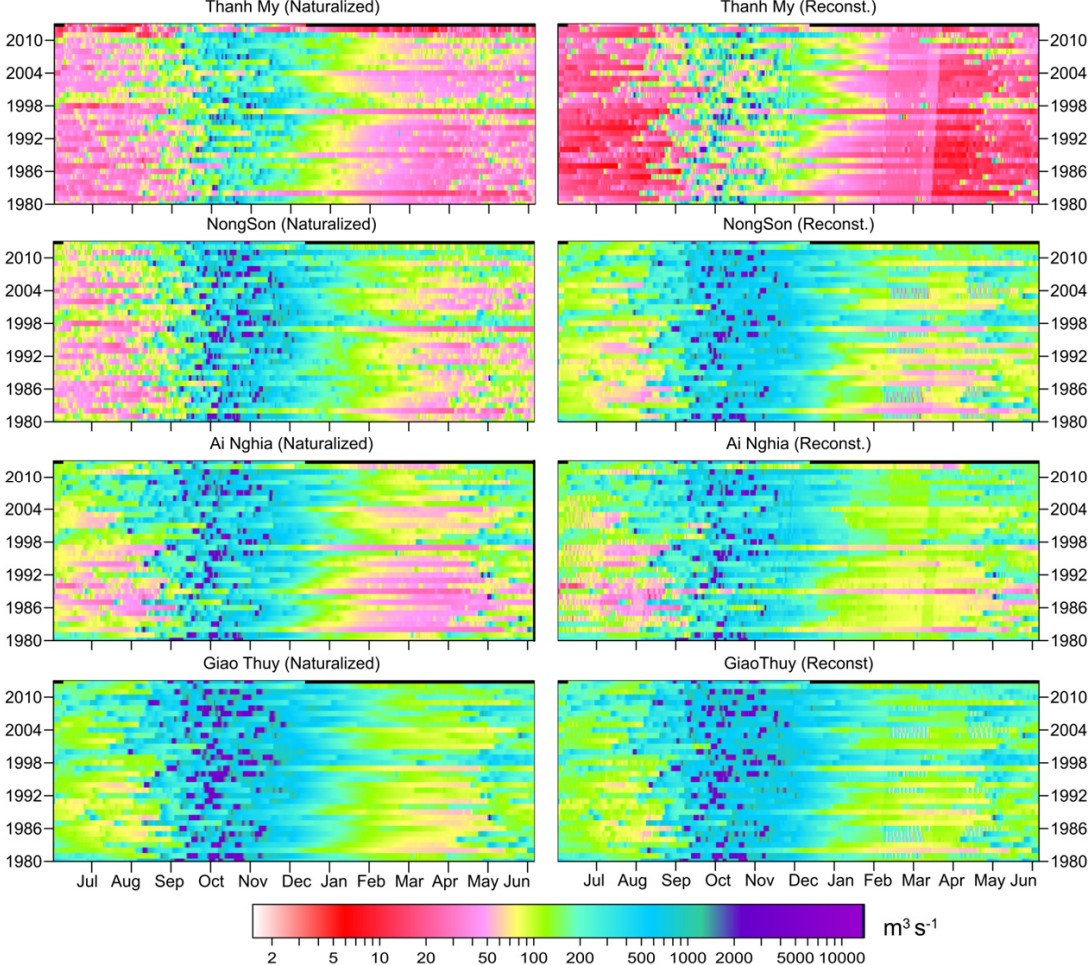

Figure 6: illustrates daily values of discharge in m³ s⁻¹ at the four discharge stations. Each pixel in the plot is one day and its colour represents discharge in m³ s⁻¹. The bottom axis represents the hydrological year, starting from July and end at June. The left side plot showing the naturalized condition based on the J2000 model simulation. The right-side figures are the reconstructed streamflow product based on the reservoir simulation model.





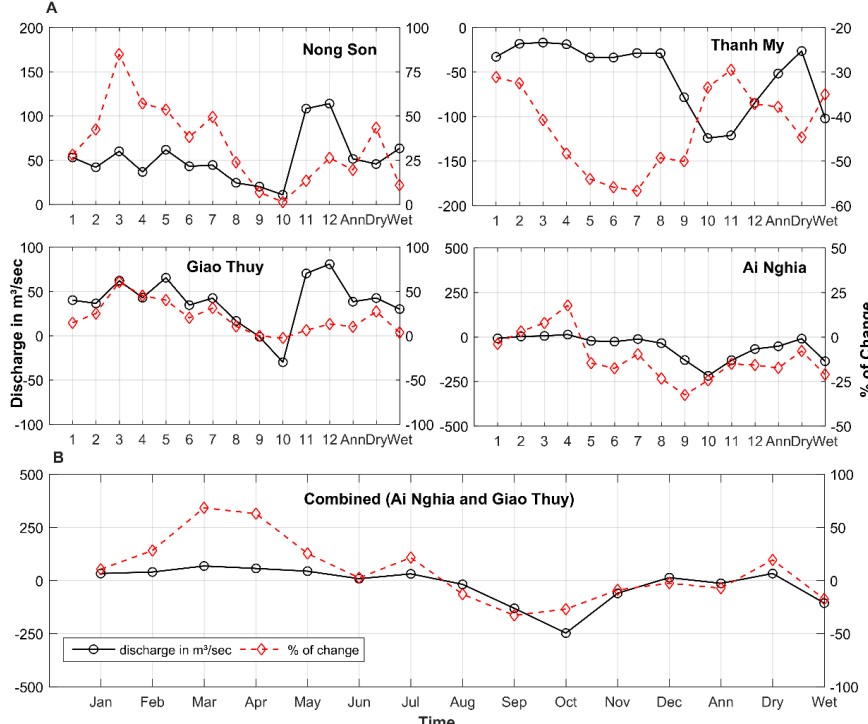

Figure 7: Reservoir impact on streamflow changes. (a) Mean differences of reconstructed streamflow pattern (Discharge in m³ s⁻¹) and the percentage (%) of changes of streamflow from the naturalized mean flow for the period of 1980-2013. The negative value indicates a decreasing flow compared with the naturalized one and vice versa. The number indicates the month starting from January. (b) Combined effect of reservoirs impact for Ai Nghia and Giao Thuy, represents the overall impact on the VGTB basin due to reservoir construction





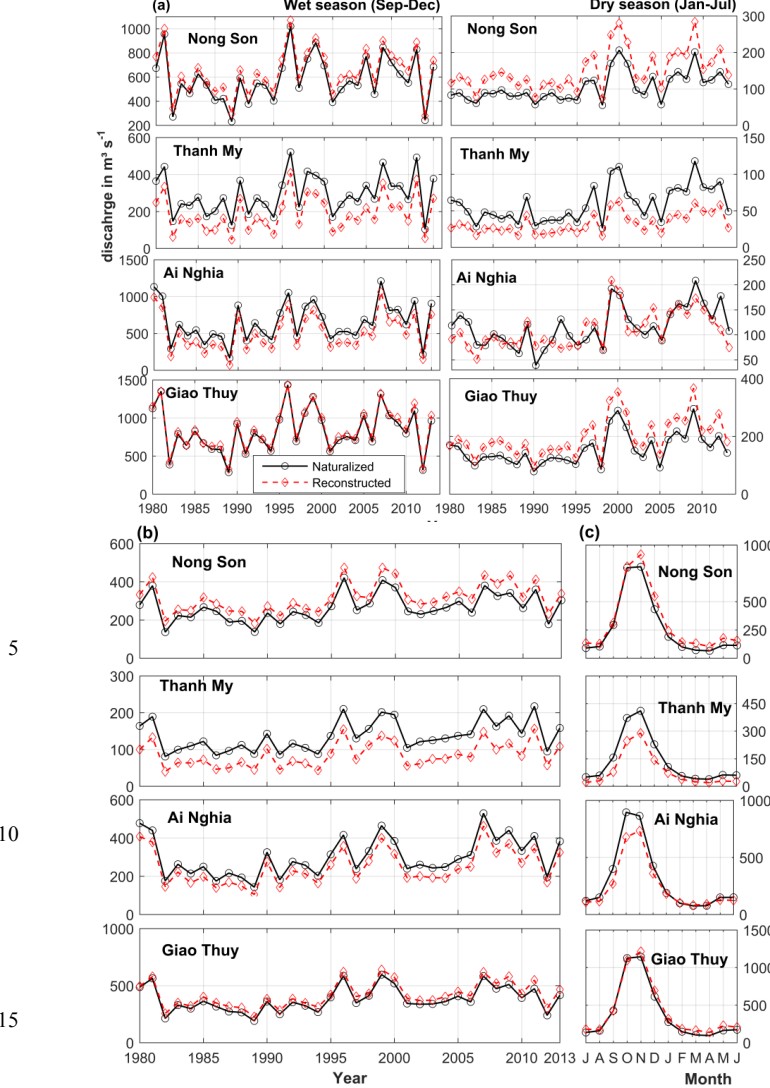

Fig 8 Comparison of mean streamflow pattern (naturalized and reconstructed streamflow); (a) comparison of mean seasonal flows for Dry (Jan to Jul) and Wet (Sep to Dec) season; (b). comparison of mean annual streamflow and; (c) comparison of mean monthly streamflow, month start from July. Units are in m$^3$ s-$^1$.

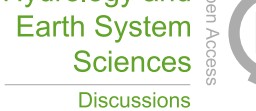

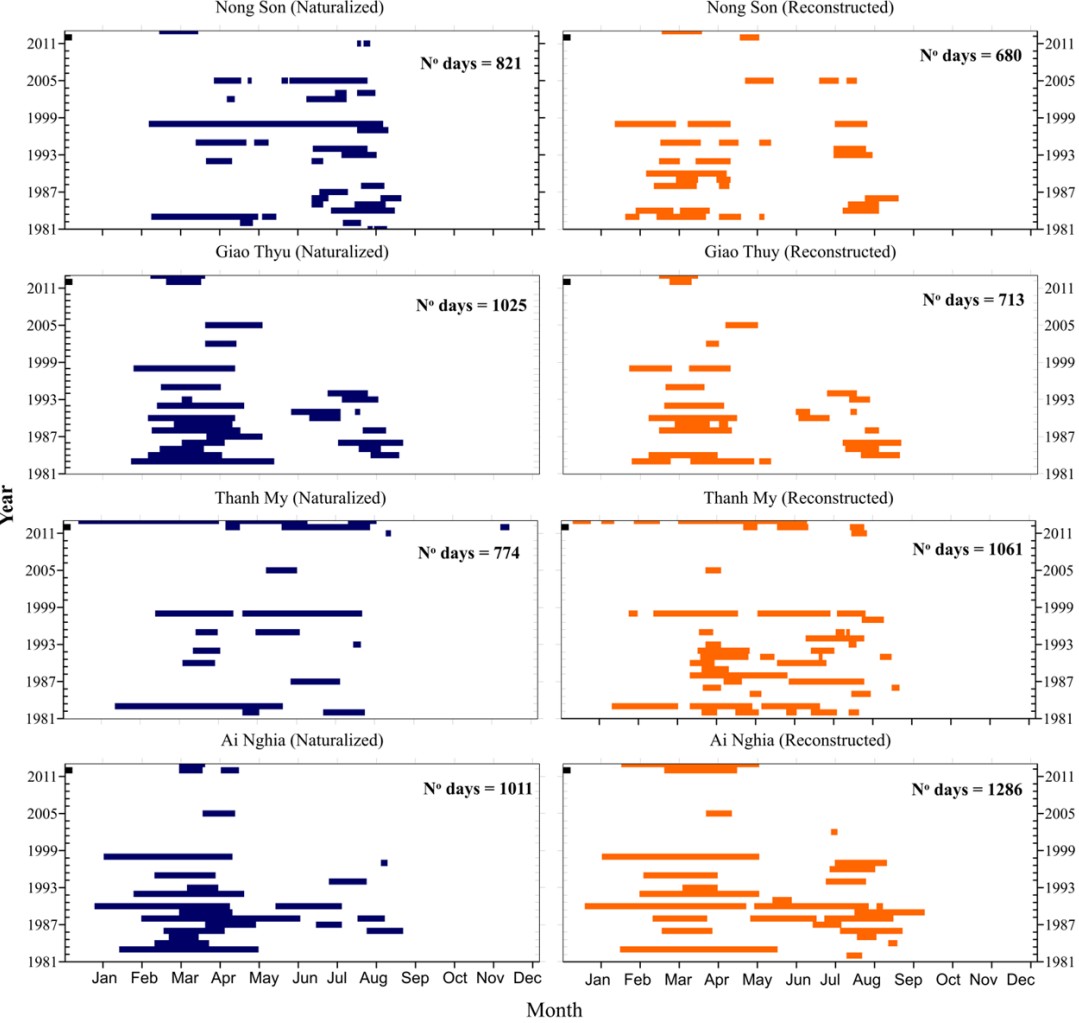

Figure 9: Number of days below the $Q_{90}$ variable drought threshold for the VGTB at the four discharge stations (1981-2013). One day of streamflow drought is a day in which the 30-day running mean discharge is below the $10^{th}$ percentile of 30-day mean discharge. The blue colour bars (left-side) show the drought onset and duration for the naturalized stream flow while the orange colour bars (right-side) represent the reconstructed reservoir impacted discharge. N0 indicates the total number of drought days.