# Peer review of "Quantifying human impacts on hydrological drought using a combined modelling approach in a tropical river basin in Central Vietnam"

_Hydrology and Earth System Sciences, 2017_

## Referee Comment (RC1) · Anonymous Referee #1 · 24 May 2017

Authors estimate reservoir impacts on hydrological drought using a catchment hydrological model combining with reservoir routing approach in a tropical river basin in Central Vietnam. The topic is interesting as it gives how extent of the reservoir operation affects seasonal variation of streamflow and thus drought occurrence in the extremely uneven distributed precipitation region. The used approaches are able to quantify the reservoir effects on streamflow. However, some conclusions need to further illustration. (1) Generally, the construction of reservoirs is to reduce the drought by smoothing streamflow variations (increase water release in the dry season and decrease the release in the flood season). However, it could shift the drought occurrence (e.g. Fig. 9). So I don't agree with authors' conclusions "we found a stronger hydrological drought

risk for the anthropogenically impacted reconstructed streamflow". (2) In the study region, one of the main effect on streamflow in the two streams is the water division from VuGia to Thu Bon. The division increases streamflow at Nong Son station and decreases streamflow at Thanh My station (Fig 7 A and 8a). So I am very interested how this water division influence drought occurrences at two streams in addition to reservoir operations. Authors need to give clear illustrations. (3) Authors only described "the reservoir should release a minimum of 25 m3s-1 water from the reservoir to the Vu Gia river (MOIT, 2011) (Page 9). How much the division amount between the two streams was used in the study? (4) The whole study is focused on the reservoir operation including water division influence on drought. So I suggest that the topic should change to be "reservoir impacts on hydrological drought. . ..". Human impacts are too broad as authors don't quantify other human influences, such as land use and land cover. (5) So in introduction, descriptions of the previous studies on modeling approaches for quantifying human activities on hydrology should be focused on mostly reservoir operations and regional water division. (6) rainfall-runoff model J2000 should be calibrated and validated by using observed streamflow discharge before reservoir operation. (7) Line 20 on Page 11: "The flow during the rainy season decreased by -2 to -38%" refers to which stream? (8) In discussion, it is not necessary to describe generally known uncertainty of the modelling. Authors can discuss uncertainty in lack of more observation data in sub-basins, e.g. calibrated parameters from one sub-basin (station) used for other sub-basins. (9) Conclusion should be revised to focus on how extent of the reservoir operation affects seasonal variation of streamflow and thus drought occurrence in the extremely uneven distributed precipitation region.

---

## Referee Comment (RC2) · Anonymous Referee #2 · 26 May 2017

**Review of the paper**

**"Quantifying human impacts on hydrological drought using a combined modelling approach in a tropical river basin in Central Vietnam"**

**by Firoz et al.**

The paper implemented a number of models and assessment method to quantify and highlight the role of reservoirs in the upper part of the change in hydrological drought in downstream of Vu Gia Thu Bon river basin (VGTB), in central Vietnam. By comparing the naturalized and reconstructed data at four discharge stations, a significant consequence of reservoir operation was found in different time scales. Not only duration and frequency, but also the severity of drought was considered with use of threshold approach. This makes the paper completely compatible with the third scoop of HESS, which aim to investigate the influence of human activity to some particular aspects including droughts. Although considering the natural- and impounded-flow has been widely used, but the successful simulation and combination of a rainfall-runoff model and a reservoir modelling based a good foundation for further study facing with the poor data observation.

**1. General comments**

The paper is quite standard structured with five main parts excluding references and shows a good smooth of writing, which helps the reader in capturing the main ideas in both title and written content. However, there are still a few concerns as presented below.

The abstract does summarize from context to method and major outcomes of the study. However, it could be more precise if the author either remove or better express the third sentence in the second paragraph without mentioning the local stakeholders.

The introduction provides a good summarized background of the topic, so that the reader can quickly obtain the wide range of application for this issue. A certain number of former researched are mentioned to strengthen the objectives. However, it would be worth if the author reveals other works in which (fully or partly) implemented the same methodologies. The objectives are clearly stated in line 17 – 26 (page 3) with a main goal and four mini-ones.

The study area is fully characterized in part 1.1 to help the reader, who are not familiar with tropical climate, catch the major identities. The status of observation data, hydropower plants and reservoirs are described in section 2.1, and are very essential to understand the circumstances in VGTB. Besides the spelling mistakes (see specific comments section), a redundancy of information is found in two parts. The author may wish to combine 1.1 and 2.1 (as suggested above) to avoid double explanation about hydrological gauges. Moreover, a few points need better coherence, for example:

- The author offers no explanation of why he chooses data set for his calculation in the period 1980–2013, whilst the discharge data available since 1976 (page 4, line 23).
- Quang Hue channel (page 5, line 15) actually diverts water from Vu Gia to Thu Bon in flood season only, thus, the author could obviously avoid this connection by explaining that this work considers the drought season, rather than assume that "Ai Nghia locate upstream of the diversion of the Quang Hue…" (line 16).
- The definition of "*Flood season*" and "*Dry season*" mentioned in page 6 (line 4) may need a sources. Otherwise, the current operation rule in VGTB defines them differently (please refer Decision 1537/QD-TTg released on 07/09/2015, Decision of Inter-reservoir operation rule in VGTB).
- Table 1 mentioned in this part is expected to use the up-to-date statistic. Since they were listed in 2008, the year of operation is not matched completely. Dak Mi4, for example, is said to start operating in 2011, but the actual activation was in 2012 which also mentioned in the results part and figure 5 later on.

- The Dak Mi 4B actually does not play important role in this work. It is not mentioned in the body of the paper, except in page 5, line 21. The author may wish to explain why it disappears in the paper, because Dak Mi 4 reflects to both Dak Mi 4A&B or only Dak Mi 4A.

The method part spreads in almost two pages which give general description about JAMS/J2000 HRU, HEC-ResSim, combined modeling and Hydrological Drought Assessment. Besides the suggestion for re-order the sub-sections (see major comments), this part could be a bit improved if:

- The performance of efficiency statistics for the J2000 is mentioned here and also provide the "significance level" if possible, rather than explain them in the result part (page 9, line 2–7). As a reader, I may question how is the goodness of E2, $R^2$.... which are shown in Table 2 and 3?
- the sub-section 2.2.4 is shortened and the reason of choosing $t_c$ = 3 days, or $z_c$ = 10 % is given. Since they are not presented in the result part, a question of whether the equations and its components are really needed to write in details?
- the definition of "*hydrological year*" (page 8, line 7) may be required to make the reader not confuse with the one "*water year*" which start at beginning of flood season. In line 11–12, it is defined as "the starting of the wet season", but in the line 11–12 (page 4), the rainy season last from September to December. The author may wish to either better distinguish them or unify one term (if they are same)
- the using of data set in each model is explained in this part, rather than in results.

The results are in an appropriate presenting, which follow sequentially the methodologies. The good point of this part is the way to deal with the data shortage, which is very common in this catchment, and they way to have long-term impact of reservoirs. I think this is very good approach. The amount of result is sufficient to the interpretation as well as compatible to the given objective. However, some sentences in this part are seen that should belong to the method- or discussion section. For instance, the explanation of how using data for model or the number of reservoirs in simulation may be better located in methodology, or the line 28–33 (page 9) should belong to discussion, and so on. There are few comments for this part as below:

- The author used data for J2000 HRU is from 1996–2005 to obtain the parameters but do not explain why that period but no former or later one.
- The Reservoir Modelling is taken for four out of eight reservoirs, but results of Song Con 2 is missing in this part, although it is shown in Fig.5
- The value of E2, $R^2$... in Table 3 may need further explanation in terms of calculation or comparison.

The paper has a very long and detail discussion with three main questions, from the applicability of the off-line coupling model to the potential uncertainties it may occur. Two limitations are discussed in this section, that makes the paper have a comprehensive view. However, it seems to me that the section 4.2 and 4.3 are more related to the technical issue, about the appropriateness of this linkage to the same issue, rather than the understanding of changes quantified. Since the title and the objective stress on quantifying human impacts on hydrological drought, I expect this will be the major part of discussion. The current argument would be helpful in a paper, which focus more on the linkage. Besides, no figure or table was mentioned in the discussion part, this would raise the question to the reader that how related the results and the aim of paper are. As pointed out above, there is some writing in results presenting discussion, thus, I think the author may wish to restructure them to make the discussion section more relevant to the objective. For example, Figure 7, 8, 9 contains the most important results to the given goal, thus, they should be discussed in this part. In addition, I would suggest to reduce section 4.2 and 4.3 if the paper is required to be shortened.

The first two sentences of the conclusion are more likely suitable for introduction rather than in conclusion. The first paragraph re-shows the methods and they are quite general, thus, it might be redundant in my view. In this step, the author may wish to relate the methods and the principal findings to help the reader have the substantial closure. I do not think that mentioning to "the reports from local stakeholders" is needed in this section, it could be better to relate to the discussion. The uncertainties expressed here in five lines making the conclusion less concise. The last paragraph shows clearly outcomes of this paper.

The literature cited is relevant to the study. I suggest to unify the order of team papers chronologically before alphabetically as guided by HESS. Furthermore, the author could also reduce the references list by choosing the ones that used for the discussion later on.

**2. Major comments**

a) Regarding to the structure: I recommend reordering a few parts. In detail, the section 1 (introduction) had sub-section 1.1, but the other 1.2 could not be found. Furthermore, since the introduction is expected to provide the literature and objectives only, the author may wish to group sub-section 1.1 and 2.1 in section 2. The methodology could either combine with the data or be a separated section. In case, the author wish to keep them as ordered, the sub-section 1.1 could join as a part of section 2.1. The results section is well presented the introduced methods consecutively, except sub-section 3.3 and 2.1.1. The author may wish to switch part 2.1.1 for 2.1.3 to make the reader easier to follow the next section. I also suggest to re-locate some parts in results (as presented above) to help the reader find easier to follow.

b) Because the author mentions in both the title and the objective that to quantify the human impacts on hydrological drought using a combined modeling approach, I expected that the impact quantified and off-line coupling are both discussed, and the former one is likely the major theme. However, in the current paper, little mention of this impact (quantity and reason) is made in the discussion. I recommend strengthening the discussion by linking to the results (figures and tables) and making it more relevant to the objective.

c) I recommend shortening the section 1.1, 2.1, 4.2 and 4.3 as explained above, to make the paper more concise.

**3. Specific comments**

The paper is written in a good expression of English. I have no objection about this issue. However, there are still some minor remarks given:

- Page 2, line 10 and 11: the double hyphens need to make sure as being necessary.
- Page 2, line 23, a comma is missing after the blanket
- Page 2, line 29: "runoff" not "run-off"
- Page 2, line 33: Wang and Hejazi (2011) not (Wang and Hejazi, 2011)
- Page 6, line 16: a double space found between "model" and "was"; line 28: "it is" not "It is"
- Page 9, line 1: data were not datawere
- Page 10, line 18: $E2 = 0.74$ or $logE2 = 0.74$
- Page 11, line 15: Thanh My not Ai Nghia
- Page 11, line 17: Fig. 7b not Fig. 7B
- Page 26, figure 2: Giao Thuy not Giao Thu
- The format should be unified. For example, many paragraphs in page 1, 13, 14, 15 and 16 have left alignment.

The paper basically follows the manuscript composition guideline (given by HESS) in terms of mathematical requirements. There are however some typical errors found in the manuscript:
- Coordinates: in page 4, line 1, coordinates of VGTB ("$6^o$ 55'–$14^o$ 55' N" not "6o 55'–14 o55' N").
- Symbols and equations:
  - page 4 and the rest of the paper: spaces must be included between number and unit, e.g. 47 % not 47%.
  - page 4, line 3: $km^2$ not km2
  - page 4, line 9: tons-ha or tons $ha^{-1}$
- Numbers: neither dots nor commas are permitted as group separators, except that the number start with the ten-thousand digit (given by HESS). Thus, 2598 not 2,598 (page 4, line 6) and so on.
- Using of hyphens (-) and en dashes (–) are quite often confused. In most cases in this paper, hyphen is used as en dash and it should be better distinguished. For example: 65-80% (page 4, line 13) should be written as 65–80 %, and so on. Please refer guideline (given by HESS) to make them correct.
- Figures and tables:
  - Figure 7 presents the percentages of changes but did not explain how this value is calculated
  - Figure 9: Giao Thuy not Giao Thyu

- Abbreviation of:
  - figures should be unified: e.g. Figure 5 (page 10, line 22) or Fig. 5 (page 9, line 24, 28)
  - letter should be first introduced. For example, MAM and JJA (page 12, line 3) are understood that March-April-May or June-July-August, but it could make confusing to the reader when first read them.

Overall, I think the off-line coupling results are considered that novel enough for publication in HESS scope. This is extremely helpful in terms of transferability to the similar river basin dealing with data shortage or poor observation network as Vietnam. However, since the linkage approach is getting more common nowadays, the paper may expect to prove some more related studies to make sure that this work more original. By the stage of publication, all the comments on this manuscript obviously need to make clear.

---

## Referee Comment (RC3) · Anonymous Referee #3 · 15 Jun 2017

**General comments**

This study presents an interesting investigation regarding the human impacts on river discharges and hydrologic droughts risks. To this end, a robust modelling approach was adopted, allowing the authors to assess the changes in streamflow caused by the construction of several reservoirs in the study area. The contribution of this paper, although relevant, is limited by a number of factors that, if addressed, could reveal a greater potential provided by the data.

From my perspective as a non-native English speaker, the manuscript is well written but the ideas need to be better presented. For instance, the reader leaves the Methods

section unaware of relevant information (model parameter, model calibration, etc) and is surprised with them in the Results section.

Although the general idea is crystal clear to me (to assess the hydrologic impacts due to the construction of dams), the means of doing so need to be clearer. Because the paper relies on three different time series (observations, naturalized and reconstructed discharges), the reader needs to understand how each one will contribute to the analysis. This could be better explained in Data and Methods, as indicated in the list in Specific Comments. Another issue is that it is not clear in Data and Methods if the naturalized discharge refers to the undisturbed discharge from 1980 until the construction of the dams or is a simulated data. There might not be enough time to address all suggestions, but there are some points that require more attention.

As far as the review criteria listed by HESS:

1. Yes, the paper address relevant scientific questions within the scope of HESS.
2. No, the paper does not present novel concepts, ideas, tools, or data.
3. No, substantial conclusions are not reached (but could).
4. Yes, the scientific methods and assumptions are valid and clearly outlined.
5. Yes, the results are sufficient to support the interpretations and conclusions.
6. No, the description of experiments and calculations are not sufficiently complete and precise to allow their reproduction by fellow scientists.
7. Yes, the authors give proper credit to related work and clearly indicate their own new/original contribution.
8. Yes, the title clearly reflect the contents of the paper.
9. Yes, the abstract provide a concise and complete summary.
10. No, the overall presentation is not well structured and clear.
11. Yes, the language is fluent and precise.
12. Yes, mathematical formulae, symbols, abbreviations, and units are correctly defined and used.
13. Yes, some parts of the paper should be clarified and moved to other sections.

14. Yes, the number and quality of references are appropriate.

15. No, the amount and quality of supplementary material are not appropriate (further information could be provided regarding the model parameters, uncertainty analysis, etc..).

Based on the relevance of the results and robust approach used, my recommendation is to publish the paper after major revision.

**Specific Comments**

*Introduction*

P4, L12: I believe this sentence is incomplete or "is" should replace "however". Please check that.

P4, L13-14: Those ranges are not clear. Almost 65 or 80 %? 70 or 85 %? Is the word "respectively" missing somewhere in this sentence? If you want to specify the range, I do not think this is the best way to do that. Please rephrase.

P4, L14: I believe a "." is missing at the end of this sentence.

P4, L15-16: How often, e.g. n times in the past y year...? Is this statement based on the author's experience or it is possible to cite someone who verified this information?

P4, L16: Please either replace "month" by "period" or "is the driest month" by "are the driest months".

*Data & Methods*

P5, L4: Are these records available online? If so, please provide an address and indicate when it was last accessed.

P5, L5: The map in Fig. 1 shows only 12 rain gauges but here it is said that 16 were considered. Please indicate the remaining gauges on the map.

P5, L15: What is the impact of such assumption?

P6, Item 2.2.1:
- it would be nice to have both periods (pristine and impacted) clearly defined here.
- which four stations? Please name them between parenthesis. It is important to understand that the reader may not be familiar with this basin and the names of the stations, dams, etc may be confusing for foreigners. It might be easier if numbers were assigned to them, e.g. DRS1 (Thanh My) and DRS2 (Nong Son), RS1 (Ai Nghia) and so on.
-There are some points that need to be made clearer. According to item 2.1.1, you have streamflow data since the 80s and the reservoirs were not constructed until late 2000s, hence $\sim$ 30 years of undisturbed streamflow data. Why, then, did you calibrate the hydrological model using only 4 years of data? Also, there is no information regarding the J2000 model calibration in Methods section.
-Am I right to assume that there is observed streamflow data regarding the period after the construction of the dams? If so, how do they compare with your "reconstructed discharge". And why weren't those observed (real) data used to assess the impact of the dams in the hydrologic regime? This is justified only in the Discussion (Pag. 13). It would be nice to have something said about that here. Regardless, a comparison between the observed and reconstructed data should be presented. Since there is no

comparison between simulated and observed discharges, how can we rely on them to assess the impacts of the dams?

P7, L3: J2000 needs data on "land use, soil, geology, . . ." It is not mentioned how these information were acquired. Model parameter description was completely overlooked.

**Results**

Although I appreciate straightforward analysis, section 3.1 is rather simplistic. Model calibration should not be done based only on statistics ($R^2$, Nash, etc. . .). I would like to see a plot comparing simulated and observed discharges and a sensitivity analysis.

Item 3.2 What are the results in the 1st paragraph? I suggest moving the proper parts to Methods and leave only those informations that concerns the reservoir modelling process.

I'm not comfortable with using the Q simulated by J2000 as reference just because "there are no gauging stations at Ai Nghia and Giao Thuy". First, if what you have at Ai Nghia and Giao Thuy are water level stations that could not be used to derive river discharge estimates because of tidal effect, how is the tidal effect accounted for in your J2000 model? If it hasn't been considered, how does that decision affect your analysis or it doesn't affect at all? Also, how far upstream the tidal has some influence? Second, I don't agree the J2000 produced "robust" results without at least seeing a Qsim vs Qobs plot. It is comprehensible that observational data availability is often an issue and, sometimes, we need to appeal to simulated data. However, the authors need to discuss the potential implications of this choice.

P9, L25: specify that these "very good agreement" refers to A Vuong reservoir.

Section 3.3: Again, some information do not belong to Results. From my point of view, only the lines after L17 report results *per se*.

P10, L26-27: This is the first time it is mentioned that the reconstructed streamflow was compared against observations. This should be explained in Methods.

P10, L27-28: This is the first time it is mentioned to which period corresponds the reconstructed streamflow (RS). Up to this point, it seemed that the RS was for the early 2010s.

*Discussion*

The authors recognize the uncertainties that need to be addressed but provides only a qualitative overview about them. It would be enlightening to know how those uncertainties affect the results. Perhaps less important (or greater) hydrologic changes would be found. These possibilities should at least be mentioned.

Section 4.2 - The authors claim that the limited rainfall data are related to the difficult access to the basin headwaters where there is no rain gauges. I wonder what could be learned from remotely sensed precipitation. Would such estimates bring more uncertainties than the regionalization methodology adopted by J2000?

P14,L1: Please cite some examples to support this claim.

P14, L33: This sentence should be in Methods.

*Conclusion*

This section should be more elaborated, showing what was learned and concluded regarding each goal listed in the Introduction. The authors could also consider renaming it to Summary (and Conclusion) as most of it is not really conclusion but a summary of the results.

The authors were too cautious in concluding the main point of this study, which is to provide evidence about the positive/negative impacts of the dams on hydrologic droughts in the study area. This should be explicitly stated here.

**Technical Corrections**

There are several problems regarding the citations. For instance, in Page 2, Line 22, it should read "Räsänen et al. (2012) quantified" instead of "(Räsänen et al., 2012) quantified". Similar issues are found throughout the manuscript: -P2, L24
- P2, L32
- P3, L20 P4, L10: extra "("
- P4, L13: extra "."
- P12, L26
- P13, L3
- P13, L18
- P13, L30
- P14, L14
-P15, L4

---

## Author Comment (AC1) · 22 Jul 2017

We thank Reviewer # 1 for the constructive feedback to our manuscript and the helpful comments which help to further improve the presentation of our findings. In the following sections, for our responses we use blue font while for reviewer's comments we use black italic font.

**Response to Reviewer #1 Comment**

*Authors estimate reservoir impacts on hydrological drought using a catchment hydrological model combining with reservoir routing approach in a tropical river basin in Central Vietnam. The topic is interesting as it gives how extent of the reservoir operation affects seasonal variation of streamflow and thus drought occurrence in the extremely uneven distributed precipitation region. The used approaches are able to quantify the reservoir effects on streamflow. However, some conclusions need to further illustration.*

*(1) Generally, the construction of reservoirs is to reduce the drought by smoothing streamflow variations (increase water release in the dry season and decrease the release in the flood season). However, it could shift the drought occurrence (e.g. Fig. 9). So I don't agree with authors' conclusions "we found a stronger hydrological drought risk for the anthropogenically impacted reconstructed streamflow".*

We regret that our argumentation has not been clearly formulated. The purpose of our study was to assess the impacts of hydropower operation and other human alterations of the hydrological system on downstream discharge. You are right and the quantified decreases in streamflow under human influence for the historical observed period do not imply a "risk" per se. However, hydropower generation at the Dak MI 4 reservoir implies a diversion from Vu Gia to Thu Bon and therefore it reduces the discharge in Vu Gia at Than My and Ai Nghia stations under hydropower operation ("reconstructed" streamflow). For the Thu Bon an increase in discharge was observed.

We now reformulated the concluding sentence: In accordance with the reports from local stakeholders, we found a stronger hydrological drought risk for the Vu Gia river supplying water to the City of Da Nang and large irrigation systems especially in the dry season. Vu Gia river experiences the most adverse effects in terms of number of drought days compared to its natural condition, with an increase of 37 % and 17 % at Thanh My and Ai Nghia station respectively.

*(2) In the study region, one of the main effect on streamflow in the two streams is the water division from VuGia to Thu Bon. The division increases streamflow at Nong Son station and decreases streamflow at Thanh My station (Fig 7 A and 8a). So I am very interested how this water division influence drought occurrences at two streams in addition to reservoir operations. Authors need to give clear illustrations.*

The diversion of the river from Vu Gia to Thu Bon at the upper part is mainly due to the construction of Dak Mi 4 dam. Although the Dam is built on the Vu Gia river, but its turbines are located at the Thu Bon catchment, therefore, any release of Dak Mi 4 through the turbine is discharged to the Thu Bon river. This lead to the increase of the discharge towards the Thu Bon

river. Therefore, any changes of the water from Vu Gia to Thu Bon is always associates with the reservoir operation, which in our case is the reconstructed streamflow.

In order to assess the drought risk, we have presented Figure 9, which shows how the number of drought days changed due to the diversion and the reservoir operation. However, we failed to properly illustrate this. Therefore, we have added one more table in the discussion section, Table 4. The results reveal that Thanh My and Ai Nghia station experienced 37.08 % and 27.20 % more drought days, while Nong Son and Giao Thuy station had a reduction in drought days of 17.17 % and 30.43 % respectively. We also found that there is a strong seasonal variation in in hydrological drought. For example, in the dry season, the streamflow is reduced almost 45 % for Thanh My, however, for Ai Nghia, this reduction is only 7.9 %. This phenomenon is mainly because of other hydropower e.g., A Vuong, Song Bung 4,5,6 and Song Con release water during the dry season for producing energy which is located other side of the Thanh My station, but contributing the flow at the Ai Nghia station. The detailed explanation will be incorporated in the revised manuscript in Section 5.1

| | | Nong Son | Giao Thuy | Thanh My | Ai Nghia |
|---|---|---|---|---|---|
| a) Drought duration (%) | | -17.17 | -30.43 | 37.08 | 27.20 |
| b) Changes of flow (%) | | | | | |
| | Ann | 19.46 | 10.09 | -37.82 | -17.41 |
| | Dry | 43.3 | 27.23 | -44.67 | -7.91 |
| | Wet | 10.84 | 3.61 | -35.03 | -21.10 |
| c) Changes of flow (in $m^3s^{-1}$) | | | | | |
| | Ann | 51.52 | 38.32 | -51.66 | -52.14 |
| | Dry | 45.65 | 42.51 | -26.43 | -9.97 |
| | Wet | 63.25 | 29.93 | -102.12 | -136.47 |

Table. 4. Impact of human alterations on drought intensity and changes of flow in the VGTB for the period 1980-2013 on an annual and seasonal scale. a) Drought duration is calculated based on percentage changes of the number of drought days from naturalized conditions to reconstructed conditions (Fig 9). b) Changes of flow (%), are calculated based on the percentage changes of the mean flow between the Naturalized and Reconstructed streamflow for the corresponding time frame. c) The changes of flow are calculated based on mean differences of reconstructed streamflow from the naturalized mean flow. Positive values indicate increasing flow or less drought intensity compared to the naturalized discharge values.

*(3) Authors only described "the reservoir should release a minimum of 25 $m^3s^{-1}$ water from the reservoir to the Vu Gia river (MOIT, 2011) (Page 9). How much the division amount between the two streams was used in the study?*

According to our observations (2011-2013), only a maximum of 12.5 $m^3s^{-1}$ is released to Vu Gia. We therefore used the actual diverted amount in this study.

*(4) The whole study is focused on the reservoir operation including water division influence on drought. So I suggest that the topic should change to be "reservoir impacts on hydrological drought: : :.". Human impacts are too broad as authors don't quantify other human influences, such as land use and land cover.*

Thank you very much for your observation and suggestions which we duly appreciate. However, we kindly ask you to leave it as "human impacts" than "reservoir impacts" given that the installation of reservoir is a human built infrastructure to serve solely human purposes such as energy generation and flood protection. Consequently, we demonstrated how such an infrastructure can influence hydrological drought. We agree that human dimension is too broad; however, it is used more as a metaphorical dimension of human activities.

*(5) So in introduction, descriptions of the previous studies on modeling approaches for quantifying human activities on hydrology should be focused on mostly reservoir operations and regional water division.*

We appreciate this comment. During our research, we evaluated previous studies regarding the possibilities to assess hydrological drought in dependence of a diversity of anthropogenic alterations. We therefore presented a comprehensive state of the art dealing with modelling and statistical approaches to assess hydrological drought. As we kindly ask you to keep it as "human impacts", therefore, we would not modified significantly the introduction section. However, we have incorporated more studies related to reservoir operation and regional water division in our introductory section in the revised manuscript.

*(6) rainfall-runoff model J2000 should be calibrated and validated by using observed streamflow discharge before reservoir operation.*

Thank you for this comment and we apologize that our explanations have not been clear. The J2000 model was calibrated and validated using observed streamflow before hydropower came into operation in 2009. The model was then used to simulate naturalized discharge.
In section 3.1, we changed the sentence: The J2000 model was calibrated and validated for the gauging station Nong Son for the period of 1996-2005 (Calibration and validation), an undisturbed period before the reservoirs were constructed in 2009.

*(7) Line 20 on Page 11: "The flow during the rainy season decreased by -2 to -38%" refers to which stream?*

Please apologize for not making this clear: here we summed the flow of Ai Nghia and Giao Thuy stations to provide an overview about the overall water availability at basin scale as shown in Fig 7b.
We changed the sentence:
The water availability in the entire basin during the wet season decreased by -2 to -38 % (Fig. 7b). This might lower the flood risk in the region.

*(8) In discussion, it is not necessary to describe generally known uncertainty of the modelling. Authors can discuss uncertainty in lack of more observation data in sub-basins, e.g. calibrated parameters from one sub-basin (station) used for other sub-basins.*

Thank you for the remarks. We agree that a description of potential uncertainty analyses is not very helpful. Our intension was to discuss potential analyses which could be carried out in a further step. Now, we have included the uncertainty analysis graph for the J2000 hydrological model as a supplementary information in the paper and include parameter estimation as well (please see our detailed response to Reviewer #3 about uncertainty and parameter estimation). In addition to this, Reviewer # 2 also raised the question about the uncertainty part (4.2 and 4.3). We therefore decided to merge and shorten sections 4.2 and 4.3.

*(9) Conclusion should be revised to focus on how extent of the reservoir operation affects seasonal variation of streamflow and thus drought occurrence in the extremely uneven distributed precipitation region.*

We have revised the conclusion focusing on how the human modified system impacted seasonal variation of streamflow and drought occurrence as suggested. However, a more detail description of this impact will be incorporated in section 5.1 as well as in the conclusion at section 6.

---

## Author Comment (AC2) · 22 Jul 2017

**Response to Reviewer #2**

We thank Reviewer # 2 for the profound evaluation of the paper and the helpful comments, which will further improve this paper. We are confident to adequately address each comment and our reply describing the planned revisions of the manuscript are highlighted in blue normal font, while the reviewer's comments are in *italic font*.

*The paper implemented a number of models and assessment method to quantify and highlight the role of reservoirs in the upper part of the change in hydrological drought in downstream of Vu Gia Thu Bon river basin (VGTB), in central Vietnam. By comparing the naturalized and reconstructed data at four discharge stations, a significant consequence of reservoir operation was found in different time scales. Not only duration and frequency, but also the severity of drought was considered with use of threshold approach. This makes the paper completely compatible with the third scoop of HESS, which aim to investigate the influence of human activity to some particular aspects including droughts. Although considering the natural- and impounded-flow has been widely used, but the successful simulation and combination of a rainfall-runoff model and a reservoir modelling based a good foundation for further study facing with the poor data observation.*

We thank you so much for the recognition of our work and related effort.

**1. *General Comments:**

*The abstract does summarize from context to method and major outcomes of the study. However, it could be more precise if the author either remove or better express the third sentence in the second paragraph without mentioning the local stakeholders.*

We agree with you that the formulation is too weak for an abstract. We reformulated the sentence also responding to reviewer # 1:

"We found a stronger hydrological drought risk for the Vu Gia river supplying water to the City of Da Nang and large irrigation systems especially in the dry season."

*The introduction provides a good summarized background of the topic, so that the reader can quickly obtain the wide range of application for this issue. A certain number of former researched are mentioned to strengthen the objectives. However, it would be worth if the author reveals other works in which (fully or partly) implemented the same methodologies. The objectives are clearly stated in line 17 – 26 (page 3) with a main goal and four mini-ones.*

We thank you very much for the comment. Although we have not found any publication in the literature which fully follows our methodologies, there are some, which partially do which we have incorporated in the introduction section. We have incorporated the work of López-Moreno et al., (2014); Wagner et al. (20017) and Wada et al. (2013).

*The study area is fully characterized in part 1.1 to help the reader, who are not familiar with tropical climate, catch the major identities. The status of observation data, hydropower plants*

*and reservoirs are described in section 2.1, and are very essential to understand the circumstances in VGTB. Besides the spelling mistakes (see specific comments section), a redundancy of information is found in two parts. The author may wish to combine 1.1 and 2.1 (as suggested above) to avoid double explanation about hydrological gauges.*

Thank you for this constructive suggestion to shorten and merge the two sections. We agreed that the repeating information is unnecessary. As per suggestion, we have merged section 1.1 under Section 2.1 and shortened the text. We furthermore included a method section as separate section 3.

*Moreover, a few points need better coherence, for example: The author offers no explanation of why he chooses data set for his calculation in the period 1980–2013, whilst the discharge data available since 1976 (page 4, line 23).*

Please apologize for not making this clear. The temperature data, which is needed for J2000 calibration, does start in 1980. Therefore, we could not use earlier data from other variables. We included an explanation in section 2.2.

*Quang Hue channel (page 5, line 15) actually diverts water from Vu Gia to Thu Bon in flood season only, thus, the author could obviously avoid this connection by explaining that this work considers the drought season, rather than assume that "Ai Nghia locate upstream of the diversion of the Quang Hue…" (line 16).*

Generally, the Quang Hue Channel in VGTB has diverted water from Vu Gia to Thu Bon all over the years, but the diversion varies between the seasons. For example, during the summer or dry season, sometimes it diverts 20% of its water while in the flood season this amount increases significantly. Please see the reference document (ICEM, 2008: PP 105). Furthermore, in the supplementary document (S1), we have included the diversion rate. Therefore, we have chosen a proxy location of Ai Nghia, to calculate the impact. In the revised text version, we will explain this diversion in detail to avoid misunderstanding.

*The definition of "Flood season" and "Dry season" mentioned in page 6 (line 4) may need a source. Otherwise, the current operation rule in VGTB defines them differently (please refer Decision 1537/QD-TTg released on 07/09/2015, Decision of Inter-reservoir operation rule in VGTB).*

Thanks for the suggestion. We have incorporated the references from the technical documents for Dak Mi 4, A Vuong hydropower operations rules. (MOIT, 2011 & 2012)

*Table 1 mentioned in this part is expected to use the up-to-date statistic. Since they were listed in 2008, the year of operation is not matched completely. Dak Mi4, for example, is said to start operating in 2011, but the actual activation was in 2012 which also mentioned in the results part and figure 5 later on.*

Thanks for the remark about the table. We have corrected and updated the table.

*The Dak Mi 4B actually does not play important role in this work. It is not mentioned in the body of the paper, except in page 5, line 21. The author may wish to explain why it disappears in the paper, because Dak Mi 4 reflects to both Dak Mi 4A&B or only Dak Mi 4A.*

Dak Mi 4B is a runoff hydropower plant, i.e., it does not have any storage functionality but uses the turbine discharge water of Dak Mi 4A to generate energy and release to Thu Bon river. Hence, we have not accounted this for model evaluation. However, as for the other hydropower reservoirs, we considered its operation in the integrated model. We will explain this in the revised manuscript.

*The method part spreads in almost two pages which give general description about JAMS/J2000 HRU, HEC-ResSim, combined modeling and Hydrological Drought Assessment.*

We thank Reviewer for the encouraging feedback.

*Besides the suggestion for re-order the sub-sections (see major comments), this part could be a bit improved if:*

*The performance of efficiency statistics for the J2000 is mentioned here and also provide the "significance level" if possible, rather than explain them in the result part (page 9, line 2–7). As a reader, I may question how is the goodness of E2, R2…. which are shown in Table 2 and 3?*

Thank you for your constructive comment. We agree to bring this into the method section 3.1. The efficiency statistics have been incorporated in the text and the evaluation scale was explained.

*the sub-section 2.2.4 is shortened and the reason of choosing tc = 3 days, or zc = 10 % is given. Since they are not presented in the result part, a question of whether the equations and its components are really needed to write in details?*

Thank you for the suggestion. We agree with the reviewer that the sub-section 2.2.4 should be shortened and that the equations need to be deleted. We intend to shorten the text by e.g., deleting the equations and associate extra text to explain this and considering the text by addressing key points how the threshold method is applied here. In the revised text we have explained why we have chosen different values for selecting the threshold level as well as the pooling process and minimum days of drought as well.

*D-3: the definition of "hydrological year" (page 8, line 7) may be required to make the reader not confuse with the one "water year" which start at beginning of flood season. In line 11–12, it is defined as "the starting of the wet season", but in the line 11–12 (page 4), the rainy season last from September to December. The author may wish to either better distinguish them or unify one term (if they are same)*

We want to thank Reviewer # 2 for these valuable comments. First, the definition "hydrological year" has been clarified in the text and we apologize for the mistake in the original manuscript. We defined that the hydrological year starts when the streams, channels or rivers are receiving water after the dry season. In VGTB, August can be referred to as the starting month, because from this month on the flow is starting to increase from its preceding month (Fig. 2). Therefore, we have defined beginning of August is the start of our hydrological year. Secondly, when considering the seasonality, September to end of December is considered as flood season and

the dry season lasts from January to August. This argument has been addressed and included in the revised manuscript .

*If the using of data set in each model is explained in this part, rather than in results. (also done in line) and the changes were made.*

Thanks for the suggestion. We understand that the using of the data set in each model should be explained in the method part. The proposed changes have been implemented in the revised manuscript

*The results are in an appropriate presenting, which follow sequentially the methodologies. The good point of this part is the way to deal with the data shortage, which is very common in this catchment, and they way to have long-term impact of reservoirs. I think this is very good approach. The amount of result is sufficient to the interpretation as well as compatible to the given objective. However, some sentences in this part are seen that should belong to the method- or discussion section. For instance, the explanation of how using data for model or the number of reservoirs in simulation may be better located in methodology, or the line 28–33 (page 9) should belong to discussion, and so on. There are few comments for this part as below:*

Thanks for these remarks and the appreciation. We fully agree with the modification proposed by the Reviewer regarding reorientation of the results into the method or discussion section. The uses of data in the model have shifted into the methodology section, for example, we have removed old section 3.1 J2000 Hydrological model calibration to simulate reservoir inflow and naturalized discharge and merged this into method section (new) 3.1. Jams/J200 HRU based Rainfall- Runoff model

*The author used data for J2000 HRU is from 1996–2005 to obtain the parameters but do not explain why that period but no former or later one.*

Thank you for this comment and we apologize that our explanations have not been clear. We have chosen this time frame because we used the observed streamflow before hydropower came into operation. In section 3.1, we changed the sentence: "The J2000 model was calibrated and validated for the gauging station Nong Son for the period of 1996-2005which was an undisturbed period before the reservoirs were constructed in 2009.

*The Reservoir Modelling is taken for four out of eight reservoirs, but results of Song Con 2 is missing in this part, although it is shown in Fig.5*

We appreciate the Reviewer comment on Song Con 2 hydropower. This has now been addressed adequately and was adopted into the revised manuscript

E-3: The value of E2, $R_2$… in Table 3 may need further explanation in terms of calculation or comparison.

Thanks for the suggestion. We have included the explanation in the results section of the hydropower modelling

*The paper has a very long and detail discussion with three main questions, from the applicability of the off-line coupling model to the potential uncertainties it may occur. Two limitations are discussed in this section, that makes the paper have a comprehensive view. However, it seems*

*to me that the section 4.2 and 4.3 are more related to the technical issue, about the appropriateness of this linkage to the same issue, rather than the understanding of changes quantified. Since the title and the objective stress on quantifying human impacts on hydrological drought, I expect this will be the major part of discussion. The current argument would be helpful in a paper, which focus more on the linkage. Besides, no figure or table was mentioned in the discussion part, this would raise the question to the reader that how related the results and the aim of paper are. As pointed out above, there is some writing in results presenting discussion, thus, I think the author may wish to restructure them to make the discussion section more relevant to the objective. For example, Figure 7, 8, 9 contains the most important results to the given goal, thus, they should be discussed in this part. In addition, I would suggest to reduce section 4.2 and 4.3 if the paper is required to be shortened.*

Thank you for this valuable suggestion. We agree that there is a lack of discussion about the key points of the paper. We therefore have changed it by including a new section 4.1 on "Quantifying human impacts" discussing the key results with reference to the figures 6 to 9. In addition to this, we have included Table 4, "Impact of human alterations on drought intensity and changes of flow in the VGTB for the period from 1980- 2013 on an annual and seasonal scale".

|  |  | Nong Son | Giao Thuy | Thanh My | Ai Nghia |
|---|---|---|---|---|---|
| a) Drought duration (%) |  | -17.17 | -30.43 | 37.08 | 27.20 |
| b) Changes of flow (%) |  |  |  |  |  |
|  | Ann | 19.46 | 10.09 | -37.82 | -17.41 |
|  | Dry | 43.3 | 27.23 | -44.67 | -7.91 |
|  | Wet | 10.84 | 3.61 | -35.03 | -21.10 |
| c) Changes of flow (in $m^3 s^{-1}$) |  |  |  |  |  |
|  | Ann | 51.52 | 38.32 | -51.66 | -52.14 |
|  | Dry | 45.65 | 42.51 | -26.43 | -9.97 |
|  | Wet | 63.25 | 29.93 | -102.12 | -136.47 |

Table. 4. Impact of human alterations on drought intensity and changes of flow in the VGTB for the period 1980-2013 on an annual and seasonal scale. a) Drought duration is calculated based on percentage changes of the number of drought days from naturalized condition to reconstructed condition (Fig 9). b) Changes of flow (%), are calculated based on the percentage changes of the mean flow between the Naturalized and Reconstructed streamflow for the corresponding time frame. c) The changes of flow are calculated based on mean differences of reconstructed streamflow from the naturalized mean flow. The positive value indicates increasing flow or drought intensity in relation to the naturalized condition.

As of Reviewer # 1, questions about the applicability of the section 4.2 and 4.3 in this research, therefore we have agreed to shorten this.

*The first two sentences of the conclusion are more likely suitable for introduction rather than in conclusion. The first paragraph re-shows the methods and they are quite general, thus, it might be redundant in my view. In this step, the author may wish to relate the methods and the principal findings to help the reader have the substantial closure. I do not think that mentioning to "the reports from local stakeholders" is needed in this section, it could be better to relate to the*

*discussion. The uncertainties expressed here in five lines making the conclusion less concise. The last paragraph shows clearly outcomes of this paper.*

We agree that repeated introductory sentences are redundant in the conclusion. We tried to follow the general guidelines how to write a conclusion by summarizing the key issues of the paper.

We now shortened these introductory sentences and also the ones referring to a potential uncertainty analysis. The updated conclusion will be included in the revised manuscript.

*The literature cited is relevant to the study. I suggest to unify the order of team papers chronologically before alphabetically as guided by HESS. Furthermore, the author could also reduce the references list by choosing the ones that used for the discussion later on.*

Thanks for the suggestions, we will follow the reference guideline of HESS. Please allow us to keep the references used for the introduction as we would like to deliver a general state of the art on how human impacts on discharge can be quantified in the scope of this paper.

**2. Major Comments-**

*Regarding to the structure: I recommend reordering a few parts. In detail, the section 1 (introduction) had sub-section 1.1, but the other 1.2 could not be found. Furthermore, since the introduction is expected to provide the literature and objectives only, the author may wish to group sub-section 1.1 and 2.1 in section 2. The methodology could either combine with the data or be a separated section. In case, the author wish to keep them as ordered, the sub-section 1.1 could join as a part of section 2.1. The results section is well presented the introduced methods consecutively, except sub-section 3.3 and 2.1.1. The author may wish to switch part 2.1.1 for 2.1.3 to make the reader easier to follow the next section. I also suggest to re-locate some parts in results (as presented above) to help the reader find easier to follow.*

We thank Reviewer #2 for the constructive feedback. We have taken them into consideration and agree to combine the sub-section 1.1 and 2.1 in section 2. Based on your suggestions, we have reorganized the paper as follows: 1. Introduction, 2. Data and Study Area, 3. Methods, 4. Results, 5. Discussion and 6. Conclusion. We have further considered to relocate some of the results in the discussion section.

*Because the author mentions in both the title and the objective that to quantify the human impacts on hydrological drought using a combined modeling approach, I expected that the impact quantified and off-line coupling are both discussed, and the former one is likely the major theme. However, in the current paper, little mention of this impact (quantity and reason) is made in the discussion. I recommend strengthening the discussion by linking to the results (figures and tables) and making it more relevant to the objective.*

Thank you for this valuable suggestion. As previously mentioned, we have updated the discussion by including a new section 4.1 on "Quantifying human impacts" discussing the key results with reference to the figures 6 - 9. In addition to this, we have included Table. 4, "Impact of human alterations on drought intensity and changes of flow in the VGTB for the period from 1980 - 2013 on an annual and seasonal scale".

*I recommend shortening the section 1.1, 2.1, 4.2 and 4.3 as explained above, to make the paper more concise.*

As explained earlier, we have merged section 1.1 and 2.1 under section 2, and 4.2 and 4.3 is shortened and merged into one section

**3. Specific comments**

*The paper is written in a good expression of English. I have no objection about this issue. However, there are still some minor remarks given:*

1. *Page 2, line 10 and 11: the double hyphens need to make sure as being necessary.*

We have changed this accordingly.

2. *Page 2, line 23, a comma is missing after the blanket*

We have accepted your comment and changes were made accordingly.

3. *Page 2, line 29: "runoff" not "run-off "*

Ans-  Thanks for the comment, change was made accordingly.

4. *Page 2, line 33: Wang and Hejazi (2011) not (Wang and Hejazi, 2011)*

Ans- Thanks for the comment, change was made accordingly.

5. *Page 6, line 16: a double space found between "model" and "was"; line 28: "it is" not "It is"*

Ans- Thanks for the comment, changes were made accordingly.

6. *Page 9, line 1: data were not datawere*

Ans- Thanks for the comment, we have changed this accordingly.

7. *Page 10, line 18: E2 = 0.74 or logE2 = 0.74*

Ans- Thanks for the remarks, It will be loge2 = 0.74, and the correction was made.

8. *Page 11, line 15: Thanh My not Ai Nghia*

Ans- Thanks for the comment, we have corrected it as Thanh My.

9. *Page 11, line 17: Fig. 7b not Fig. 7B*

Ans- Thanks for the comment, we have changed it as Fig. 7b.

10. *Page 26, figure 2: Giao Thuy not Giao Thu*

Ans- Thanks for the comment, We have changed as Giao Thuy.

> 11. The format should be unified. For example, many paragraphs in page 1, 13, 14, 15 and 16 have left alignment.

Ans- We have corrected the formatting for the mentioned pages.

*The paper basically follows the manuscript composition guideline (given by HESS) in terms of mathematical requirements. There are however some typical errors found in the manuscript:*

> a) *Coordinates: in page 4, line 1, coordinates of VGTB ("6o 55'–14o 55' N" not "6° 55'–14 °55' N").*

Ans: Thanks for the comments, we have corrected this as suggested.

> b) *page 4 and the rest of the paper: spaces must be included between number and unit, e.g. 47 % not 47%.*

Ans: Thanks for the comment, we have corrected all the number and units, as suggested in the text.

> c) *page 4, line 3: km2 not km²*

Ans- Thanks for the comment. We have changed it accordingly.

> d) *page 4, line 9: tons-ha or tons ha$^{-1}$*

Ans- Thanks for the comment. We have changed it accordingly.

> e) *Numbers: neither dots nor commas are permitted as group separators, except that the number start with the ten-thousand digit (given by HESS). Thus, 2598 not 2,598 (page 4, line 6) and so on.*

Ans: Thanks for the comment, we have corrected all the numbers as suggested.

> f) *Using of hyphens (-) and en dashes (–) are quite often confused. In most cases in this paper, hyphen is used as en dash and it should be better distinguished. For example: 65-80% (page 4, line 13) should be written as 65–80 %, and so on. Please refer guideline (given by HESS) to make them correct.*

Ans: Thanks for the remarks and observations. We have changed hyphens (-) to en dashes (-) as suggested.

*Figure and Tables:*

> g) *Figure 7 presents the percentages of changes but did not explain how this value is calculated*

Ans- Thanks for the comment. We agreed that it needs a bit more explanation in addition what we have explained in the text. So the correction we have made as follows:

The percentage of changes of flow is calculated based on the percentage changes of the mean flow between the Naturalized and Reconstructed streamflow for the corresponding time frame. We have incorporated this in to the Figure 7a.

h) *Figure 9: Giao Thuy not Giao Thyu*

Ans- Thanks for the correction. We have changed this as Giao Thuy

*Abbreviation of:*

a) *figures should be unified: e.g. Figure 5 (page 10, line 22) or Fig. 5 (page 9, line 24, 28)*

Ans: Thanks for the suggestions. We have changed it as Fig. 5 and the guideline has followed for the rest of the manuscript.

b) *letter should be first introduced. For example, MAM and JJA (page 12, line 3) are understood that March-April-May or June-July-August, but it could make confusing to the reader when first read them.*

Ans: Thanks for the remarks. We have introduced to the abbreviated letters in the revised manuscript.

*Overall, I think the off-line coupling results are considered that novel enough for publication in HESS scope. This is extremely helpful in terms of transferability to the similar river basin dealing with data shortage or poor observation network as Vietnam. However, since the linkage approach is getting more common nowadays, the paper may expect to prove some more related studies to make sure that this work more original. By the stage of publication, all the comments on this manuscript obviously need to make clear.*

We thank you so much for the recognition of our work. To address your suggestions, we have included some more recent literature, showing the coupling approach to evaluate the reservoir impact on the streamflow changes. However, there is no literature which exactly follows our approach. Below is the list of the new references that we incorporated into the revised manuscript.

López-Moreno, J. I., Vicente-Serrano, S. M., Beguería, S., García-Ruiz, J. M., Portela, M. M., and Almeida, A. B.: Dam effects on droughts magnitude and duration in a transboundary basin: The Lower River Tagus, Spain and Portugal, Water Resour. Res., 45, 6, doi:10.1029/2008WR007198, 2009.

Wagner, T., Themeßl, M., Schüppel, A., Gobiet, A., Stigler, H., and Birk, S.: Impacts of climate change on stream flow and hydro power generation in the Alpine region, Environ Earth Sci, 76, 33, doi:10.1007/s12665-016-6318-6, 2017.

Wada, Y., van Beek, L. P. H., Wanders, N., and Bierkens, M. F. P.: Human water consumption intensifies hydrological drought worldwide, Environ. Res. Lett., 8, 34036, doi:10.1088/1748-9326/8/3/034036, 2013.

**Other references**

ICEM: Strategic Environmental Assessment of the Quang Nam Province Hydropower Plan for the Vu Gia-Thu Bon River Basin, Prepared for the ADB, MONRE, MOITT & EVN, Hanoi, Viet Nam, 205 pp., 2008.

MOIT: Decision Number 6801/QD-BCT, Decision on Dak Mi 4 Reservoir Operation, Ministry of Investment and Trade, Socialist Republic of Vietnam, Hanoi, Viet Nam, 2011

MOIT: Decision Number 1997/QD-BCT, Decision on A Vuong Reservoir Operation, Ministry of Investment and Trade, Socialist Republic of Vietnam, Hanoi, Viet Nam, 2012

---

## Author Comment (AC3) · 22 Jul 2017

**Response to Reviewer # 3**

We thank Reviewer # 3 for the detailed evaluation of the paper and the helpful comments, which will further improve this paper. We are confident to adequately address each comment and our reply are highlighted in blue normal font, while the reviewer's comments are in *italic font*.

**General comments**

*This study presents an interesting investigation regarding the human impacts on river discharges and hydrologic droughts risks. To this end, a robust modelling approach was adopted, allowing the authors to assess the changes in streamflow caused by the construction of several reservoirs in the study area. The contribution of this paper, although relevant, is limited by a number of factors that, if addressed, could reveal a greater potential provided by the data.*

Thank you for your appreciation of our effort. We have addressed all of your comments in our revised manuscript.

*From my perspective as a non-native English speaker, the manuscript is well written but the ideas need to be better presented. For instance, the reader leaves the Methods section unaware of relevant information (model parameter, model calibration, etc) and is surprised with them in the Results section.*

Answer: Thank you for this suggestion! We will include more information about the model parameters, calibration procedure in the method section. We have reorganized the Data and Methods section as follows: instead of presenting the Data and Methods in one section, we have separated them: 2. Data, 3. Methods. A detailed description about the hydrological model has been included in the Methods section which incorporates the calibration procedure, the parameter estimation and model efficiency statistics. In addition to this, a detailed explanation of the model can be found in Fink et al. (2013) and Nepal et al. (2014).

*Although the general idea is crystal clear to me (to assess the hydrologic impacts due to the construction of dams), the means of doing so need to be clearer. Because the paper relies on three different time series (observations, naturalized and reconstructed discharges), the reader needs to understand how each one will contribute to the analysis. This could be better explained in Data and Methods, as indicated in the list in Specific Comments. Another issue is that it is not clear in Data and Methods if the naturalized discharge refers to the undisturbed discharge from 1980 until the construction of the dams or is a simulated data. There might not be enough time to address all suggestions, but there are some points that require more attention.*

Answer: We appreciate the comments. The definition of "naturalized" and "reconstructed" discharges were described in method section 2.2.1 (p 6, line 14 – 24). For our drought risk assessment, we simulated the "naturalized" and "reconstructed" discharge to be able to evaluate the changes of streamflow due to reservoir construction. These simulated time series were calibrated against the observed period. The naturalized discharge is the output from the J2000 hydrological model, which simulates discharge for pristine conditions without the intervention of the reservoirs, while the "reconstructed" discharge is the output of the reservoir simulation model, that accounts for the hydropower operation influences in the streamflow for the same time from 1980 – 2013. In our analysis, observed data – referred to as the measured discharge data at the stations are only used for the calibration and the evaluation of the simulated results.

**Specific Comments**

**Introduction**

*P4, L12: I believe this sentence is incomplete or "is" should replace "however". Please check that.*

We have changes the sentence accordingly, "The climate in the VGTB basin is characterized by a strong rainy season lasting from September to December."

*P4, L13-14: Those ranges are not clear. Almost 65 or 80 %? 70 or 85 %? Is the word "respectively" missing somewhere in this sentence? If you want to specify the range, I do not think this is the best way to do that. Please rephrase.*

Thank you for the suggestion. We have rephrased the sentences as follows:

Rainfall during the wet season accounts for 65 to 80 % of the total annual rainfall, with 40 to 50 % of the annual rainfall occurring in October and November regularly causing severe floods (Souvignet et al., 2013).

*P4, L14: I believe a "." is missing at the end of this sentence.*

We have corrected it and put a ".".

*P4, L15-16: How often, e.g. n times in the past y year: : :? Is this statement based on the author's experience or it is possible to cite someone who verified this information?*

Thanks for the comment. We have changed the sentences as suggested.

"The extended dry season lasts from January to August and is frequently accompanied by droughts (e.g., in year 1982, 1983, 1988, 1990, 1998, 2005, 2012 and 2013)" Nauditt et al.,2017

*P4, L16: Please either replace "month" by "period" or "is the driest month" by "are the driest months".*

The sentence is corrected as suggested.

"February to April are the driest months, accounting for only 3–5 % of the total annual rainfall, resulting in severe water shortages and problems with saline intrusion at the coast (Souvignet et al., 2013)"

**Data & Methods**

*P5, L4: Are these records available online? If so, please provide an address and indicate when it was last accessed.*

Answer: Sorry for not making this clear. These data are not freely available and were bought them in the scope of a BMBF funded research project (www.lucci-vietnam.info). We now included a sentence in the data section about this as well as the missing acknowledgements.

*P5, L5: The map in Fig. 1 shows only 12 rain gauges but here it is said that 16 were considered. Please indicate the remaining gauges on the map.*

Thank you for this hint. We have updated the map (see below), which shows the location of 17 rain gauge stations and the text has been corrected in the manuscript.

[Figure]

**Figure-1: Map of the Study Area**

*P5, L15: What is the impact of such assumption?*

Answer: We agree with the reviewer that this was not clearly explained. The flow diverted through the Quang Hue channel is strongly dependent on tidal changes and seasonal variation in streamflow of both Thu Bon and Vu Gia. Also, the impact is minor compared to the reservoir operation generated discharge from the tributaries. We did not incorporate the dynamics of the channel as there are no daily data on diversion flow. Our results also show that the contribution from tributaries to Vu Gia downstream of Thanh My are more relevant for the discharge at Ai Nghia. We included such an explanation to make this clear.

The following text is incorporated in the hydro meteorological data section in the revised manuscript.

There is no control mechanism how to release the exact amount of water (see Table S1, the routing rules of water diversion by MoNRE) diverted from Vu Gia to Thu Bon through Quang Hue channel. Hence, we in our study we assumed that Ai Nghia station is located upstream of the diversion of the Quang Hue channel to avoid complexity. We found that the proxy station can be well represented to capture the influences of reservoir impact on the downstream without having any potential errors, as it accounts for the overall water balance. It further helps to avoid any associate uncertainty due to the diversion which however is quite dynamic and difficult to predict without rigorous monitoring campaigns.

*P7, L3: J2000 needs data on "land use, soil, geology, …:" It is not mentioned how these information were acquired. Model parameter description was completely overlooked.*

Answer: Thank you for your comment on the data issues for the J2000 model. We have not mentioned this in the original text as the data used for the J2000 model were described in Fink et al., 2013.

Soils: Soil map (1:100,000) (National Institute of Agriculture Planning and Protection, 2005) + 150 soil profile descriptions in the catchment to derive soil-model parameters for the various soil classes

described in the map. Geology map (1:100,000) (Department of Geology and Minerals of Vietnam, Hanoi, 1997). Land-cover classification of Landsat images for the year 2010 (Avitabile et al,. 2016). The digital elevation model (DEM) was derived from contour lines and points from a digital map (scale 1:50,000) using the topography to-raster algorithm of ArcGIS. The resulted DEM had a resolution (cell size) of 25 m.

The following text has been incorporated-

A detailed description of the spatial (e.g., soil, vegetation, digital elevation model, land use and geology) and hydro-climatic data used for hydrological model was described in Fink et al., (2013, p 1828).

Model parameters and calibration procedure were not presented as the focus of this article was mainly to show the drought risk assessment based on the coupled modelling approach. These are described in Fink et al., (2013) and Nepal et al., (2014) and can also be found under http://jams.uni-jena.de/documentation/. However, we will include the parameter estimation values as supplementary material in the revised manuscript.

Table 1: Parameters selected for the model calibration (other parameters of the model were left to default values during calibration)

| Calibrated parameters | Short description | calibrated Value | Range |
|---|---|---|---|
| soilMaxDPS | Maximum depression storage capacity | 2 | 1.0 - 5 |
| soilMaxInfSummer | Maximum infiltration in summer | 40 | 1 -200 |
| soilMaxInfWinter | Maximum infiltration in winter | 100 | 1 -200 |
| soilDistMPSLPS | MPS/LPS distribution coefficient | 0.68 | 0 - 1 |
| soilDiffMPSLPS | MPS/LPS diffusion coefficient | 0.4 | 0 - 1 |
| soilConcRD1 | Recession coefficient for overland flow | 1.2 | 1.0 -3.0 |
| soilConcRD2 | Recession coefficient for interflow | 3.5 | 2.0 - 10 |
| soilPolRed | Potential reduction coeffiecient for aET computation | 3 | 1.0 - 10 |
| soilMaxPerc | Maximum percolation rate | 20 | 1.0 - 20 |
| gwRG1Fact | Adaptation of the fast groundwater outflow | 1 | 0.1 - 10 |
| gwRG2Fact | Adaptation of the baseflow | 0.4 | 0.1 - 10 |
| gwRG1RG2dist | RG1-RG2 distribution coefficient | 0.5 | 0 - 1 |
| flowRouteTA | River routing coefficient | 10 | 1 -100 |

The parameters that were calibrated are affecting three domains of the model. The most important ones are governing the simulation of the soil processes. The soilMaxDPS governs how much water can be hold back on the soil surface before surface runoff occurs. The  soilMaxInfSummer and soilMaxInfWinter are there to influence the maximum infiltration in the dry and rainy season. The soil characteristics in the model are described by a dual porosity approach were the large pores (excess water) and the medium pores (usable field capacity) are represented in two different storages (MPS and LPS). The parameter soilDistMPSLPS is influencing the distribution of infiltrated water between LPS and MPS. SoilDiffMPSLPS affecting the diffusion from LPS to MPS. The recession coefficients soilConcRD1, soilConcRD2 are influencing the travel time of the runoff components surface runoff and interflow. The reduction of actual evapotranspiration (actET) to potential evapotranspiration (potET) is influenced by the soilPolRed which is a shape parameter for the actET, potET function according to the actual soil moisture conditions. The Groundwater runoff components (fast groundwater and base flow) influence the two recession adaption coefficients gwRG1Fact and gwRG2Fact. The distribution between these two components is influenced by the gwRG1RG2dist distribution coefficient. The recession in the river network is affecting the simulated recession in the river network.

**Results**

*Although I appreciate straightforward analysis, section 3.1 is rather simplistic. Model calibration should not be done based only on statistics (R2, Nash, etc: : :). I would like to see a plot comparing simulated and observed discharges and a sensitivity analysis.*

Answer: Thank you for this suggestion. Due to the high number of figures presented in the paper we could not show the hydrograph simulation performance. We therefore include the following figures to this response:

[Figure]

Figure 2: Hydrograph simulation compared to observed discharge (1996-2005) for the Thu Bon river (Nong Son station)

[Figure]

Figure 3: Hydrograph simulation compared to observed discharge (1996-2005) for the Vu Gia river at Thanh My station. Also please refer to Fink et al (2013).

Sensitivity Analysis of the calibrated parameters

[Figure]

**Figure 4: Sensitivity of calibrated parameters for the Thu Bon catchment (gauge Nong Son) with the Nash–Sutcliffe efficiency criterion.**

Based on 1000 Monte-Carlo simulation we estimated the sensitivity of the parameters used for calibration. The method performed for this sensitivity analyses is the "regional sensitivity analysis" (RSA) (Hornberger and Spear, 1981) utilizing the Nash–Sutcliffe efficiency criterion. This method estimates the impact of a parameter and its interactions with model outputs (Nepal et al. 2012). The results in Figure 1 shows the importance of the different calibrated parameters for the simulated runoff according to the Nash–Sutcliffe efficiency criterion. This example shows that the parameters with the highest sensitivity (infiltration parameters, surface runoff coefficient and river runoff coefficient) are parameters which have their main influence on quick runoff components and peak flow. In contrast to that, the parameters which affect the overall water balance are less important (e.g. SoilDiffMPSLPS, SoilDistMPSLPS, soilPolRed). One reason is the use of the Nash–Sutcliffe efficiency with is focusing on the high flows, another the extreme water surplus in the rainy season where the evapotranspiration plays only a minor role.

*Item 3.2 What are the results in the 1st paragraph? I suggest moving the proper parts to Methods and leave only the information that concerns the reservoir modelling process.*

Answer: thank you for this suggestion. We agree that this part belongs to the Methods section and shifted it to in the method section.

*I'm not comfortable with using the Q simulated by J2000 as reference just because "there are no gauging stations at Ai Nghia and Giao Thuy". First, if what you have at Ai Nghia and Giao Thuy are water level stations that could not be used to derive river discharge estimates because of tidal effect, how is the tidal effect accounted for in your J2000 model? If it hasn't been considered, how does that decision affect your analysis or it doesn't affect at all? Also, how far upstream the tidal has some influence?*

Answer: Thank you for these comments. The data of Ai Nghia and Giao Thuy stations show the following key constraints:

1. Ai Nghia station is subject to flooding during the rainy season, therefore, discharge during the rainy season cannot be used with the given rating curve.
2. The Giao Thuy station is influenced by tidal waves (although it is located 38 km upstream from the sea), therefore the water level data of Giao Thuy station cannot be accurately presented by a rating curve.
3. The most important constraint is the dynamic water diversion. As explained previously, there is no control mechanism which measures the exact amount of water diverted from Vu Gia to

Thu Bon through Quang Hue channel. It is therefore difficult to predict how much water is diverted without long term measurements. This leads to an increased uncertainty in the water level data.

We agree that the model cannot consider hourly tidal effects. Our purpose was to assess drought risk at this point in dependence of upstream climatic, human and catchment related circumstances. We assessed water availability at the daily, monthly, seasonal and yearly timescale for the irrigation system entrance. Therefore, hourly tidal information will not affect the results of this study.

*Second, I don't agree the J2000 produced "robust" results without at least seeing a Qsim vs Qobs plot. It is comprehensible that observational data availability is often an issue and, sometimes, we need to appeal to simulated data. However, the authors need to discuss the potential implications of this choice.*

Answer: we agree with your suggestion and incorporated a section on uncertainties related to these simulations in the discussion. Below (in the Discussion section) you find a Qsim versus Qobs plot including an uncertainty band.

*P9, L25: specify that these "very good agreement" refers to A Vuong reservoir.*

Answer: thank you for this hint. We have included the name of A Vuong.

*Section 3.3: Again, some information do not belong to Results. From my point of view, only the lines after L17 report results per se.*

Answer: thank you for this suggestion. We agree that this part also belongs to the Methods section and shortened and shifted it to methods section.

*P10, L26-27: This is the first time it is mentioned that the reconstructed streamflow was compared against observations. This should be explained in Methods.*

We agree with you and explained this in the method sections.

*P10, L27-28: This is the first time it is mentioned to which period corresponds the reconstructed streamflow (RS). Up to this point, it seemed that the RS was for the early 2010s.*

Answer: We agree with the reviewer, we did not explain in the methods section that the models were calibrated against the observed daily streamflow available for the years 1980 to 2013. As the reservoirs were constructed after 2009, we needed to use the "pristine" streamflow to calibrate J2000 and the reservoir impacted streamflow after 2009 to calibrate HEC RESSIM. We have incorporated this explanation to the Methods section.

**Discussion**

*The authors recognize the uncertainties that need to be addressed but provides only a qualitative overview about them. It would be enlightening to know how those uncertainties affect the results. Perhaps less important (or greater) hydrologic changes would be found. These possibilities should at least be mentioned.*

Answer: We agree with your comment and we now mentioned this in the discussion. Fink did an uncertainty analysis with the Nong Son data using 1000 model runs. Figure 5 shows the 5 % best simulations in the grey shaded area; the blue line indicates the measurements. The blue line represents the observed discharge and moves within the uncertainty band which is an indicator for a robust modelling. The graph shows the largest uncertainty during high flows and the recession phases.

[Figure]

**Figure 5: 5 % best simulations (range grey shaded) versus observed discharge (blue line) at Nong Son.**

*Section 4.2 - The authors claim that the limited rainfall data are related to the difficult access to the basin headwaters where there is no rain gauges. I wonder what could be learned from remotely sensed precipitation. Would such estimates bring more uncertainties than the regionalization methodology adopted by J2000?*

Answer: Thank you for this suggestion. Yes indeed in some cases satellite based rainfall estimates do perform better in closing the water balance than observed data. Since we wanted to analyse long term effects we needed longer time series as the station data which were available from 1980 till 2013In the scope of this study we have not considered satellite based rainfall estimation products as they are only available for shorter periods (Zambrano-Bigiarini et al., 2017).

*P14, L1: Please cite some examples to support this claim.*

…..R2 of the regression line is used to determine if the relation between the rain and the altitude is strong enough to be used for the modelling. A threshold value of R2 of 0.75 is typically used for this decision (Krause 2001, Nepal 2012, Biskop 2016).

*P14, L33: This sentence should be in Methods.*

Answer: Thanks for the comment, we shifted it to the Methods section.

**Conclusion**

*This section should be more elaborated, showing what was learned and concluded regarding each goal listed in the Introduction. The authors could also consider renaming it to Summary (and Conclusion) as most of it is not really conclusion but a summary of the results.*

*The authors were too cautious in concluding the main point of this study, which is to provide evidence about the positive/negative impacts of the dams on hydrologic droughts in the study area. This should be explicitly stated here.*

Answer: We agree that the conclusion is not containing the key findings in terms of the objectives. We updated the conclusion accordingly in the revised manuscript.

**Technical Corrections**

There are several problems regarding the citations. For instance, in Page 2, Line 22, it should read "Räsänen et al. (2012) quantified" instead of "(Räsänen et al., 2012) quantified". Similar issues are found throughout the manuscript:

-P2, L24

We have corrected it as "Räsänen et al. (2012) quantified"

- P2, L32

we have corrected it as "Wang and Hejazi (2011)"

- P3, L20

We have deleted extra "(" in the text

- P4, L10: extra "("

We have deleted extra "(" in the text

- P4, L13: extra "."

We have deleted extra "." in the text

- P12, L26

We have corrected it as "Adam et al. (2007)"

- P13, L3

We have corrected it as "Nauditt et al. (2017)"

- P13, L18

We have corrected it as "are described e.g. in Walker et al. (2003); Refsgaard et al. (2007); Beven and Binley (1992)."

- P13, L30

We have corrected it as "Krause (2002)"

- P14, L14

We have corrected it as "Mateus and Tullos (2016)"

-P15, L4

We have corrected it as "by Nepal et al. (2014)"

**References:**

Avitabile, V., Schultz, M., Herold, N., Bruin, S. de, Pratihast, A. K., Manh, C. P., Quang, H. V., and Herold, M.: Carbon emissions from land cover change in Central Vietnam, Carbon Management, 7, 333–346, doi:10.1080/17583004.2016.1254009, 2016.

Biskop S.: Advancing the understanding of hydro-climatic controls on water balance and lake-level variability in the Tibetan Plateau - Hydrological modeling in data-scarce lake basins integrating multi-source data. PhD Thesis, Friedrich Schiller University of Jena, 2016.

Fink, M., Fischer, N., Führer, N., Firoz, A., Viet, T., Laux, P., and Flügel, W.-A.: Distributive hydrological modeling of a monsoon dominated river system in central Vietnam, in: MODSIM2013, 20th International Congress on Modelling and Simulation, Piantadosi, J., Anderssen, R., and Boland, J. (Eds.), MODSIM2013, 20th International Congress on Modelling and Simulation, Sydney, Australia, 1826–1832, 2013.

Hornberger GM, Spear RC.: An approach to the preliminary analysis of environmental systems. Journal of Environmental Management12:7–18, 1981.

Krause P.: Das hydrologische Modellsystem J2000: Beschreibung und Anwendung in grossen Flusseinzugsgebieten. Schriften des Forschungszentrum Jülich. Reihe Umwelt/Environment; Band 29, 2001.

Nauditt, A., Firoz, A., Viet, T. Q., Fink, M., Stolpe, H., and Ribbe, L.: Hydrological drought risk assessment in an anthropogenically impacted tropical catchment, in: Land Use and Climate Change Interactions in Central Vietnam: LUCCi, Nauditt, A., and Ribbe, L. (Eds.), Water Resources Management and Development, Springer Book Series, 2017.

Nepal S.: Evaluating upstream–downstream linkages of hydrological dynamics in the Himalayan region. PhD Thesis, Friedrich Schiller University of Jena, 2012.

Nepal, S., Krause, P., Flügel, W.-A., Fink, M., and Fischer, C.: Understanding the hydrological system dynamics of a glaciated alpine catchment in the Himalayan region using the J2000 hydrological model, Hydrol. Process., 28, 1329–1344, doi:10.1002/hyp.9627, 2014.

Souvignet, M., Laux, P., Freer, J., Cloke, H., Thinh, D. Q., Thuc, T., Cullmann, J., Nauditt, A., Flügel, W.-A., Kunstmann, H., and Ribbe, L.: Recent climatic trends and linkages to river discharge in Central Vietnam, Hydrol. Process., 28, 1587–1601, doi:10.1002/hyp.9693, 2013.

Zambrano-Bigiarini, M., Nauditt, A., Birkel, C., Verbist, K., and Ribbe, L.: Temporal and spatial evaluation of satellite-based rainfall estimates across the complex topographical and climatic gradients of Chile, Hydrol. Earth Syst. Sci., 21, 1295–1320, doi:10.5194/hess-21-1295-2017, 2017.